# Global water level variability observed after the Hunga Tonga-Hunga Ha'apai volcanic tsunami of 2022

Adam T. Devlin[1,2,3,4], David A. Jay[5], Stefan Talke[6], Jiayi Pan[1,4,7,*]

[1]School of Geography and Environment, Jiangxi Normal University; Nanchang, Jiangxi, China

[2]Cooperative Institute for Marine and Atmospheric Research, School of Ocean and Earth Science and Technology, University of Hawai'i at Mānoa; Honolulu, HI, United States of America

[3]Department of Oceanography, University of Hawai'i at Mānoa; Honolulu, HI, United States of America

[4]Institute of Space and Earth Information Science, Chinese University of Hong Kong; Shatin, Hong Kong, China

[5]Department of Civil and Environmental Engineering, Portland State University; Portland, OR, United States of America

[6]Department of Civil and Environmental Engineering, California Polytechnic State University; San Luis Obispo, CA, United States of America

[7]Key Laboratory of Poyang Lake Wetland and Watershed Research of Ministry of Education; Nanchang, China

*Correspondence to*: Jiayi Pan (panj@cuhk.edu.hk)

**Abstract** The eruption of the Hunga Tonga-Hunga Ha'apai volcano on Jan 15[th] of 2022 provided a rare
opportunity to understand global tsunami impacts of explosive volcanism and to evaluate future hazards, including
dangers from "volcanic meteotsunamis" (VMTs) induced by the atmospheric shock waves which followed the
eruption. The propagation of the volcanic and marine tsunamis was analyzed using globally-distributed 1-min
measurements of air pressure and water level (from both tide gauges and deep-water buoys). The marine tsunami
propagated primarily throughout the Pacific, reaching nearly 2.0 m at some locations, though most Pacific
locations recorded maximums lower than 1.0 m. However, the VMT resulting from the atmospheric shock wave
arrived before the marine tsunami and propagated globally, producing water-level perturbations in the Indian
Ocean, the Mediterranean, and the Caribbean. The resulting water level response of many Pacific Rim gauges was
amplified, likely related to wave interaction with bathymetry. The meteotsunami repeatedly boosted tsunami wave
energy as it circled the planet several times. In some locations, the VMT was amplified by as much as 35-fold
relative to inverse barometer, due to near-Proudman resonance and topographic effects. Thus, a meteotsunami
from a larger eruption (such as the Krakatoa eruption of 1883) could yield atmospheric pressure changes of 10mb
to 30mb, yielding a 3-10m near-field tsunami that would occur in advance of (usually) larger marine tsunami
waves, posing additional hazards to local populations. Present tsunami warning systems do not consider this threat.

## 1. Introduction

The immense energy of the Hunga Tonga-Hunga Ha'apai volcanic eruption (20.54°S, 175.38°W) at 0415 UTC on 15 January 2022 (hereafter the "Tonga Event") was one of the strongest eruptions of the past 30 years (Witze, 2022). It produced a variety of atmospheric waves at various levels that travelled the globe multiple times (Adam, 2022; Duncombe, 2022). Lamb waves were produced first from the eruption (Lamb, 1911; Nishida et al., 2014). These travel with a celerity $V \sim 310$ ms$^{-1}$, which is faster than marine gravity longwaves, except in the deepest parts of the ocean. Lamb-wave generation is driven by a complex process involving eruption-generated pulses of pressure, temperature, and density gradients in the atmosphere. The Tonga Event induced Lamb waves and closely-following atmospheric gravity waves which were detectable up to the ionosphere (Lin et al., 2022; Wright et al., 2022; Themmens et al., 2022; Kulichkov et al., 2022; Matoza et al., 2022; Kubota et al., 2022; Nishida et al., 2014). Closer to the surface, the pressure pulse added momentum to the ocean surface through a pressure-gradient forcing that pushed the ocean surface in the direction of the positive-pressure gradient (Lynett et al., 2022). The subsequent and slower atmospheric gravity waves had phase speeds of 200-220 ms$^{-1}$, about the speed of long waves in the deep ocean. Recent work has confirmed the presence of a slower internal Perkeris wave (Perkeris, 1937; 1939) which has helped resolve long-standing issues about atmospheric resonance (Watanabe et al., 2022).

The Tonga Event differed from previously observed tsunamis, with unexpected dynamic atmospheric variability in addition to the expected oceanic variability. The most closely related historical corollary is the Krakatoa Event of 1883, which had much stronger atmospheric shock waves and yielded global water level fluctuations, due to a stronger volcanic meteotsunami (VMT) than occurred after the Tonga Event. Krakatoa also differed from the Tonga Event, because the former event was land-based, while the latter was due to eruption of a submarine volcano whose apex was between 500 and 1000m below the ocean surface. This layer of water likely shielded and contained much of the explosive impact of the eruption; if the same event happened at sea level, it would likely have been much more destructive. The Tonga Event is thought to have generated waves via multiple mechanisms: air-sea coupling from the shockwave in the immediate vicinity, collapse of the underwater cavity after the explosion (which controlled near-field impacts), and air-sea coupling with the pressure pulse that circled the Earth and was responsible for the VMT (Lynett et al., 2022).

The unusual nature of the Tonga Event has inspired a plethora of publications. Several observation-based studies documented and cataloged the initial dynamics of the eruption (Yuen et al., 2022; Poli and Shapiro, 2022), the propagation of the atmospheric shock wave, its record-setting volcanic plume height (e.g., Carr et al., 2022),

the impacts of the marine tsunami in the Pacific, and the far-field water level fluctuations distant from the Pacific
that were due to the VMT (e.g., Carvajal et al., 2022).

Ocean-atmospheric interactions due to the Tonga Event produced far-field water-level perturbations

comparable to those from the 2004 Sumatra (Titov et al., 2005), the 2010 Chile (Rabinovich et al., 2013), and the
2011 Tohoku Events (Mori et al., 2011). It spread throughout the Pacific Ocean and was measured in all ocean
basins except the Arctic. Regional studies documented the VMT impacts to water level on the Russian coasts of
the Sea of Japan (Zaytsev et al., 2022), as well as along the coasts of Mexico. The Gulf Coast of Mexico was only
affected by the VMT, while the Pacific coast was impacted by both the marine tsunami and the VMT (Ramírez-
Herrera et al., 2022).  Tsunami signatures were also seen in parts of the South China Sea, such as Lingding Bay
near Hong Kong (Wang et al., 2023).

Tsunami characteristics around Japan were closely studied, due in part to an extensive array of ocean

bottom pressure instrumentation (S-net and DONET) established after the Tohoku megathrust earthquake of 2011
(Tanioka, 2020; Kubo et al., 2022; Kubota et al., 2021).  The directionality, velocity, and intensity of the tsunami
were estimated through array analysis of this data network, finding that the amplitude of the first tsunami waves
diminished upon reaching shallow water regions, and that the wave split after passing the continental shelf
(Yamada et al., 2022). Different pressure sensors recorded different velocities, because they were located in
different water depths (Kubo et al., 2022).

Several studies have approached the Tonga Event through numerical modelling (e.g., Heidazadeh et al.,

2022; Kubo et al., 2022; Kubota et al., 2022; Tanioka et al., 2022; Sekizawa et al., 2022; Saito et al., 2022).
Typical tsunami models do not include pressure terms in the shallow-water equations, because atmospheric effects
are usually small for seismic tsunamis (Yeh et al., 2008), however, the pressure terms are vital for a meteotsunami.
Accordingly, Gusman et al. (2022) employed a simplified air wave model to generate oceanic waves in a tsunami
model. This model showed that ocean waves are excited by the passage of the air wave, and this generation is
more effective over oceanic trenches. Also, repeated passes of the air wave slowed the decay of the tsunami.

The global extent and unusual nature of the Tonga event provides a unique opportunity to investigate the

dynamics and impacts of a volcanic tsunami, especially the VMT component.  The worldwide network of high-
frequency, real-time water level (WL) stations and other instrumentation improved significantly after the Sumatra
and Tohoku tsunamis, allowing for detailed study of how sensitive different locations and geometries are to
volcanically-induced atmospheric perturbations.  Though severe devastation during the Tonga Event was confined

to the immediate vicinity (mainly at other Tongan islands; see e.g., Lynett et al, 2022), most Pacific observation systems remained operational. Using these records, we assess the global spatial and temporal patterns of the tsunami and show that significant WL variations were produced in distant locations, primarily due to Lamb waves. Our investigation of 308 tide gauges where the tsunami could be detected (nearly 1000 locations were screened), 30 deep-water buoys, and 137 air pressure stations shows a patchwork of amplification, with some locations highly susceptible to meteotsunami impacts and others relatively insensitive.

We document here how the VMT was induced after the passage of the atmospheric shockwave(s) before the marine component, ahead of tsunami forecasts (where they were available) and occurred in areas where the marine tsunami was absent. We will address the following questions in this work:

- What is the amplification potential of these waves, as observed by the unprecedented number of gauges now available?
- Could a more significant volcanic event, such as a VEI 6 or 7 eruption, cause a VMT of dangerous proportions ahead of forecasted arrival times, and in areas not reached by marine tsunami waves?
- How does the persistence of a VMT under repeated passes of a planetary-scale shockwave over many days contribute to overall water levels?

## 2. Meteotsunami background

Tsunamis of volcanic origin are uncommon; less than 150 have been documented (Levin and Nosov, 2009), and aside from a few large events like Krakatoa (Wharton, 1888), most have had only local or regional footprints. Volcanic tsunamis can occur when magma rapidly displaces water, and major eruptions such as the Tonga Event can drive a planet-circling atmospheric shockwave that induces water level fluctuations globally. Volcanic activity is not, however, the only source of atmospheric tsunamis – local atmospheric disturbances can cause "meteotsunamis", independent of seismic or volcanic activity (Šepić et al., 2014; Šepić et al, 2015; Olabarrieta et al., 2017; Monserrat et al., 2006; Ripepe et al., 2016; Vilibic et al., 2016). Such meteotsunamis may have amplitudes up to 3-5m and can cause significant coastal damage. Some meteotsunami events can be deadly, such as the 1954 meteotsunami of Lake Michigan which led to the drowning of seven fisherman in Chicago (Press, 1956). Meteotsunamis are a common occurrence in the Black and Mediterranean Seas (e.g., Vilibic and Sepic, 2009), Australia, the Persian Gulf (e.g., Heidarzadeh et al., 2020), the Great Lakes of North America, and perhaps other, less documented locations. Meteotsunamis can even occur during good weather, as they can be forced by

far-field atmospheric disturbances. A wealth of information about the history and dynamics of meteotsunamis can
be found in Rabinovich (2020).
The water level fluctuations induced worldwide by atmospheric waves after the Tonga Event are a form
of meteotsunami, using "meteo" in its larger context as referring to phenomena of the atmosphere in general, and
not just weather. VMTs and weather-driven meteotsunamis share similar physical dynamics, but with several
important distinctions. First, weather-related meteotsunamis move more slowly than VMTs, meaning that
resonance with ocean waves occurs at shallower depths. Second, since weather-related meteotsunamis have a
purely atmospheric origin, they may allow some predictability via observations of weather conditions, whereas
meteotsunamis generated by an eruption such as the Tonga Event happen with less warning. Third, weather-
related meteotsunamis, while potentially destructive, are most often singular events, and do not typically have
multiple instances within a short period, such as what was seen with the Tonga Event and the repeating "ringing"
of water levels for each pass of the atmospheric shockwave. Fourth, weather-related meteotsunamis will typically
only impact discrete locations or regions, whereas the Tonga Event impacted sites worldwide. Finally, the periods
or frequencies of the forcing events (weather-related vs volcanic) are also likely distinct from one another, which
may imply different responses at any particular harbor.
VMTs are generated by a combination of Lamb and Perkaris waves that result from atmospheric
explosions like Krakatoa or the Tonga Event which move, in this case, at ~1115km hr$^{-1}$ (see Methods and
Appendix A), while weather-related meteotsunamis are driven by strong, but slower weather disturbances (Šepić
et al, 2015). The importance of this difference can be explained in terms of Froude number, $F_A$:
$$F_A = \frac{V}{\sqrt{gH}}, \tag{1}$$

where: $V$ is the atmospheric disturbance speed, $H$ is water depth, and $g$ is gravitational acceleration. For a VMT,
$F_A > 1$ for almost the entire ocean, while resonant, near-critical, conditions ($F_A \sim 1$) occur at moderate ocean
depths for meteo-tsunamis.
Atmospheric forcing of tsunamis has been analyzed in linear (Garret, 1976) and more realistic nonlinear
contexts (Pelinovsky et al., 2001). In either case, the solution consists of a forced ocean wave moving with the
atmospheric disturbance, plus forward and backward free waves. Shallow water, linear free waves of small
amplitude have celerity $c \approx \sqrt{gH}$, while nonlinear theory, relevant for $F_A \geq 1$, yields dispersive waves. The
forced wave has amplitude proportional to $\frac{V^2}{V^2-c^2}\Delta P_A$ (13), with a "nominal amplification" relative to an inverse
barometer effect of $a_n = \frac{V^2}{V^2 - c^2}$; $\Delta P_A$ is the $P_A$ (air pressure) disturbance; $a_n > 1$ for most of the open ocean.
When $F_A \sim 1$, the forced and forward-moving free waves coalesce, and the atmosphere feeds energy into the ocean
(Proudman resonance), allowing waves to grow linearly with fetch (Williams et al., 2021). The actual forced wave
"amplification factor," $\alpha$, observed at an ocean bottom pressure gauge depends on many factors and may differ
from $a_n$.

For a subcritical wave, a *rise* in $P_A$ of 1mb causes a *fall* in WL of 10mm via the inverse barometer effect.

However, VMT-forced waves are supercritical in ocean depths <9.7km, and the Bernoulli effect causes a *positive*
$P_A$ spike to drive a forced marine wave as a *rise* in WL (Garret, 1976) with Proudman resonance occurring only
in the deepest ocean waters. Amplification disappears ($a_n \cong 1$) in shallow water, but interaction of the forced
wave with the continental slope and shelf will energize the free waves, allowing shallow-water amplification
(Garret, 1976). A VMT differs from a weather-related meteotsunami in that strong amplification is limited to deep
ocean trenches, compensated by a potential for $\Delta P_A$ to be larger in the VMT case than for the weather-related case.
We define the overall amplification of a tsunami at a tide gauge, encompassing Proudman resonance and local
effects, $\beta$.

What happens when a forced VMT wave encounters a sudden change in depth? A depth change, from

deep to shallow, requires the forced wave amplification, $a_n$, to decrease towards unity because $V^2 \gg c^2$ on the
shallow side, spawning transmitted and reflected waves. The transmission and reflection coefficients defined by
Garret (1976) suggest that the wave transmitted onshore as a VMT which approaches from the ocean side will be
considerably larger than the wave reflected back to the coast, as a VMT moves offshore. The offshore-directed
case is also different in that the forced wave must be small, because the shelf will typically be less than a
wavelength wide and the fetch for its development is limited. These factors suggest that coastal amplitudes may
be different for the direct and antipodal approaches of a VMT to any given location. While Garret's formulae
strictly apply to transitions that are abrupt (i.e., occur over a distance small relative to a VMT wavelength of ~180
to 1100km). they still provide approximate guidance for VMT interactions with the continental shelf.
The dynamics at sharp, but more complex features, like deep ocean trenches, is presumably something
intermediate between the Proudman resonance case, where the forced wave amplification factor, $a_n$, adjusts as
the wave propagates, and the fission of the forced wave into transmitted and reflected components. Also, at a
trench near the coast, the depth difference will typically be larger on the landward side than on the seaward side,
driving a larger transmitted wave. The transmitted wave may further grow over a continental shelf landward of
the trench as $h^{-\frac{1}{4}}$, per Green's law (Green, 1838). Other resonance processes may occur in specific circumstances.
Pattiaratchi and Wijeratne (2015) cite quarterwave resonance and Greenspan resonance. Both of these processes
have specific geometric requirements, and the large velocity of VMT waves renders both of these mechanisms
less likely for a VMT than for weather-related events. Finally, the propagation of the atmospheric shockwave
may also be influenced by atmospheric temperature gradients (Amores et al., 2022), which may in turn modulate
the marine response to the shockwave.

**3. Methods**

*3.1 Data Inventory*

We employ high-frequency (1-min) water level (WL) data from multiple worldwide data sources,

including coastal tide gauges and deep-water pressure buoys (see Appendix A for detailed procedures and
uncertainty estimates). Air pressure ($P_A$) data at a variety of temporal resolutions (1, 6, and 10 min) were also
acquired. Some regions, such as the European Atlantic coast, the East China Sea, and the Arctic Ocean did not
show any tsunami-like WL fluctuations. In addition, some locations (e.g., Spain) that might have registered a
tsunami lacked data during the relevant period. The buoys provide 1-min data during "active" WL events and 15-
min data otherwise. However, many were not triggered until the atmospheric shockwave had passed; thus, the
resultant VMT was often not captured, though the marine tsunami signal was clearly observed. In total, data from
308 tide gauges (out of ~1000 investigated) and 30 (out of ~60) deep-water buoys are employed, with 210
locations in the Pacific, and 98 in the rest of the world. Metadata for all tide gauges and deep-water buoys analyzed
in this study (latitude, longitude, data source, and distance from the Tonga volcano) are given in Table S1, and
metadata for air pressure stations are given in Table S2. We also list the tide gauges that were investigated but not
analyzed in Table S3, along with the reason for not using them, and show a color-coded map of the unanalyzed
locations in Figure S1. We use detrended residual WLs to quantify the amplitudes of the largest positive and
negative tsunami wave amplitudes at all stations from January 14 to 20, 2022. We also apply an EEMD analysis
(Huang et al., 1998) to all WL and $P_A$ data to remove low frequency components and biases in mean water level
to yield data in which the tsunami-related signals are dominant.

*3.2 Water Level (WL) Analysis*

VMT magnitudes and arrival times, and the amplitudes of the largest positive and tsunami waves at each

location, were determined from the WL residuals via numerical and visual estimation of the residual time series
(see Appendix A for details of calculations and a discussion of inherent uncertainty in this study). We compare
the distances and "first arrival" times at all tide gauges stations via robust regression (Holland and Welsch, 1977)
to estimate VMT celerity. MATLAB continuous wavelet transform (CWT; Rioul and Vetterli, 1991; Torrence
and Compo, 1998; Lilly, 2017) routines are applied to the WL and $P_A$ residuals to confirm approximate arrival
times (accurate within half a filter length) and to investigate the frequency response at each location. These are
discussed for selected locations. $P_A$ data (onshore and offshore) and are compared with WL variability to
investigate the relative synchronization of the $P_A$-spikes and associated WL variability. This is performed at
certain Pacific locations, as well as in the Caribbean and Mediterranean Sea regions, where observed WL
variations are solely due to atmospheric effects.
### 3.3 Energy Decay Analysis and β factor calculations
We calculate the energy decay of the Tonga event and compare to other recent tsunamis. Following
Rabinovich, (1997) and Rabinovich et al (2013), we detide 1-min NOAA WL data, remove any residual trend,
and then produce power spectra for 6hr segments of the WL residual, with an overlap of 3 hours between
successive analyses. A multi-tapered method (McCoy et al., 1998) was applied, because it reduces noise and edge
effects, but still conserves energy. The energy within the tsunami band (between 10 minutes and 3 hours) was
then integrated for each 6hr period and an exponential decay model of form $E = E_o e^{\frac{-t}{t_d}}$ applied, where $E_o$ is the
peak energy in the fit and $t_d$ is the e-folding (decay) time scale.
We use the $P_A$-spike and the related WL fluctuation amplitudes to estimate β at locations where the VMT
was observed and where co-located or nearby $P_A$ records were available. β is calculated as the ratio of the
maximum (positive) residual WL at VMT arrival divided by the maximum (positive) air pressure spike, with $P_A$
converted to a WL level fluctuation assuming the usual inverted barometer effect of 10mm WL change for 1mb
$P_A$ change. In total, we are able to calculate β at 231 locations. For the "first arrival" of the VMT, we only consider
waves arriving on 15 January, but for the β calculations, we use the largest WL amplitude closely following a $P_A$-
spike visible in the record; for many locations in the Atlantic and Mediterranean, this occurred on the second or
third pass of the atmospheric disturbance (Jan 16[th]).
## 4. Results
### 4.1. Global tsunami impacts as determined from tide gauges
The Tonga Event produced a VMT with a global footprint, along with a marine tsunami confined
primarily to the Pacific (Figure 1). VMT-related perturbations were recorded along the west coast of Africa, in

the Mediterranean and Caribbean Seas, in the Indian Ocean, and elsewhere (Fig. 1(a),(c)). Tsunami arrival times at most places closely correlate with arrival of atmospheric waves (Fig. 1(b),(d)), which propagated concentrically from the source around the planet, reconverging at the antipode. See also Tables S4-S6 and Figures S3-S12.

The largest amplitude far-field WLs from the marine tsunami occurred at dispersed Pacific Ocean locations, without a clear spatial pattern (Fig. 1(a),(b)). Several gauges within 3000 km of the eruption registered tsunamis >1m. Moderate tsunamis were measured at most island locations. In Hawaii, only Kahului measured waves >0.5 m; several islands in French Polynesia also reached this level. Consistently stronger responses occurred around the periphery of the Pacific, with wave heights of >1m at Kushimoto, Japan, four locations in Chile, four locations in California, and one in Alaska. Away from Tonga, the largest maximum and minimum measured WLs in the Pacific occurred at Chañaral, Chile (+1.73m and -1.95m); the largest in the US was Port San Luis, CA, at +1.34m. A ~2m tsunami was reported, but not measured, near Lima (https://www.nytimes.com/ 2022/01/21/world/americas/peru-oil-spill-tonga-tsunami.html). VMT amplitudes are small (<0.1m) in most locations (Fig. 1(d)), moderate (up to 0.15m) at certain locations in Chile, the Northeastern Pacific, Russia, and Hawai'i, and up to 0.22 m at some locations in Japan, Australia, and New Zealand (Table S5).

The "first arrival" map (Fig. 1(c)) shows a circular pattern emanating outwards from Tonga. Robust regression between the VMT first-arrival times and the distances from Tonga yield a slope of $1115\pm3$ km hr$^{-1}$ (Figure S2), about 90% of the sound speed at sea-level (1225 km hr$^{-1}$), and similar to the estimate of 1080-1170 km hr$^{-1}$ for the Krakatoa tsunami (Garret, 1976). Estimates from tide-gauge arrivals yield a smaller VMT celerity estimate ($1054\pm7$km hr$^{-1}$; Figure S2(b)), because the waves observed at tide gauges are subcritical, free waves that fall behind the Lamb waves in coastal waters. A similar regression analysis gives a celerity estimate of $708\pm8$ km hr$^{-1}$ for the marine tsunami wave, consistent with a mean ocean depth of about 5km. In the Pacific, the fairly regular VMT arrival pattern can be contrasted with the less regular arrival times of the largest maximum/minimum amplitude marine tsunami (Fig. 1(b)) and the time difference between "first arrival" and highest water level (Figure S9 and S10). The latter emphasizes that the VMT can occur some hours before the marine tsunami, where both were observed.

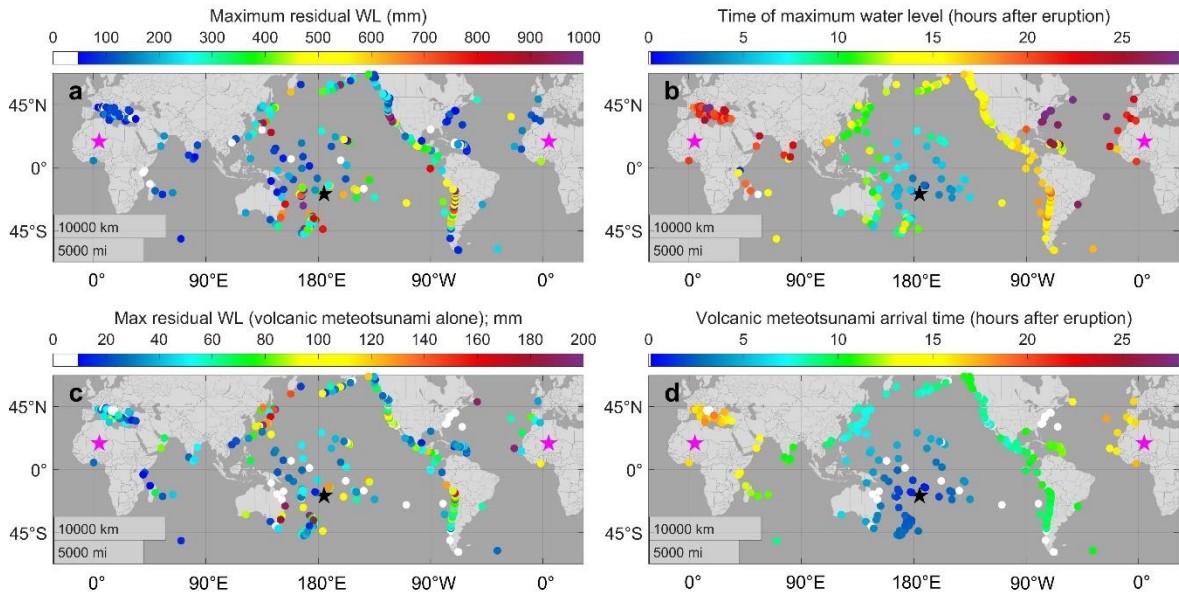

**Figure 1.** Tonga tsunami global manifestations: (a) maximum amplitude of combined volcanic (VMT) and marine tsunamis; (b) time of maximum amplitude; (c) first arrival VMT amplitude; and (d) VMT arrival time. White markers in (c) and (d) indicate locations where meteotsunami properties could not be determined. The location of the eruption and its antipode are shown by black and magenta stars, respectively. Maps made in MATLAB using data from Natural Earth.

Several Indian Ocean tide gauges (East Africa, Oman, Sri Lanka, and India) show WL changes shortly
after the atmospheric waves arrived, but display little evidence of a marine tsunami. In the Atlantic Ocean there
was a strong signal in Senegal, Ghana, and in the Cabo Verde, Canary, and Azores Islands. The Azores showed a
large WL amplitude (~0.6m), but this area is undergoing volcanic activity with frequent seismicity. While no
nearby air pressure record is available to confirm a relationship to the meteotsunami wave in the Azores, no strong
seismic activity was recorded either; hence, the causality of this result is uncertain. All of these gauges are located
within ~3000 km of the antipode of the Tonga Event (20.54° N, 4.62° E in the Sahara Desert), where the concentric
shock waves re-converge.  The resulting interference pattern may have increased the magnitude of atmospheric
waves in some places and the subsequent VMT, and masked others.
In the Eastern North Atlantic, small tsunamis occurred after the second pass of the VMT wave on 16
January, e.g., at St. Johns, Canada (~0.2m).  Storminess after 16 January precluded further detection there and in
the Baltic Sea; and little or no signal was seen on the European Atlantic Coast at any time. Wide-spread VMTs
occurred in the Caribbean and Mediterranean Seas, the latter being close to the antipodal point of the shockwave.
In both regions, successive occurrences of the VMT wave have different impacts on WL variability.
These results suggest that VMT characteristics vary between closely spaced stations, because of local
bathymetry, ambient currents, and the orientation relative to the source (Šepić et al., 2015; Garett, 1976; Williams
et al., 2021). VMT properties also change with atmospheric stratification and due to dispersion as the shockwave
propagates; the directionality of the VMT (towards or from land) also matters (Garett, 1976). Thus, the level of
response from a VMT event is locally variable, despite its global reach.
*4.2. Tsunami propagation in the Pacific as determined from deep water buoys*
The network of the National Data Buoy Center (NDBC) deep-water tsunami warning buoys provides
significant spatial coverage of the Pacific and can reveal the offshore characteristics of strong oceanic signals like
tsunamis (e.g., surface amplitude) without contamination by surface swell. These buoys generally provide a 15-
min temporal resolution but, when "triggered" by large signals, record 1-min data. We examined all available
buoys but found that many buoys did not record any data at all during the Tonga event. Thirty NDBC buoys in
the Pacific recorded at least part of the marine tsunami; however, only a subset caught the VMT (12 buoys).
Locations are given in Figure 2(a) and details of the buoys are given in Table S1.  Ten locations measured the
VMT in the Western Pacific, one in Alaska, one in Hawaii, and none in the Eastern Pacific. The Western Pacific
data reveals a similar spike-like waveform, with a steep rise followed by a rapid decrease. The magnitude of the
VMT-induced WL response is nearly consistent across the basin, except at two of the nearest buoys to Tonga
(55015 and 51425), where amplitudes were larger, 70 and 58mm, respectively. All other VMT magnitudes were
between 25 and 40mm, independent of distance from Tonga (Figure 2(b)).
The energy generated by the Tonga tsunami may have been sustained by repeated returns of the
atmospheric wave at many locations. Can the spatial characteristics of energy decay be suggested from the limited
buoy data? We next make an estimate of the "persistence" of the tsunami in the Pacific by determining the length
of time (in hours) that the buoys were "triggered" in each region of the Pacific for one-minute resolution
observations. This metric, possibly influenced by instrumental noise (or gauge problems) at some locations, allows
a simple, if imperfect, estimate of tsunami energy decay for individual buoys and for regional averages. We omit
buoy 52406 (which recorded at high resolution for > 30 hr, for reasons unclear) and determine a median regional
"persistence" in the southwest Pacific (i.e., the buoys nearest to Tonga) of 9hr, while the buoys immediately west
of Tonga had a median regional persistence of 6.5 hr. At the periphery of the Pacific, the median regional
persistence was 610 hours in the Northwestern Pacific (Japan and surrounding areas), 9 hours in the Northern
Pacific (Alaska), 10 hours in the Northeastern Pacific (California-Canada), and 13 hours around South America.
Thus, we generally see a longer persistence in far-field Pacific regions than in near-field regions (Figure 2(c)).
The maximum VMT magnitude (where detected) and the persistence times at all buoys are given in Table S7.
A subset of five buoys provides an effective summary of the VMT behavior in deep water (Figure 2(d)).
Two buoys (52402 and 21420) are close to being a great circle with each other and the Tonga eruption; buoy
52402 is ~ 5000 km from Tonga, while 21420 is ~2700 km further towards the southern coast of Japan. The VMT
maximum WL at the first buoy is about 38mm versus 30mm at the second; the subsequent WL oscillations at both
buoys are similar in form. This suggests that the VMT response of the marine WL decayed very slowly, at least
across the Pacific basin. The full set of WL responses at all buoys are given in the Supplement and compared by
region (Figures S13-S18).

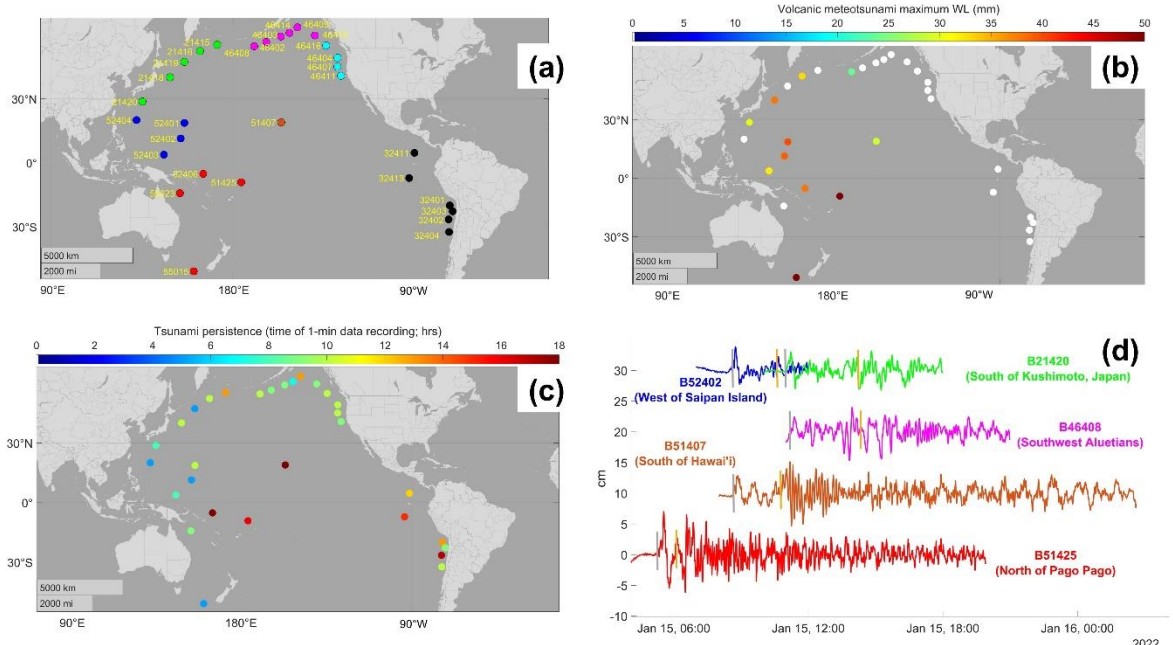


**Figure 2** Pacific deep-water NDBC buoys used to detect the VMT and marine tsunami of the Tonga event. (a)
Buoy locations and NDBC buoy designation numbers (Table S1), with colors used to show Pacific regional
delineation (red: Southwest; orange: Central; dark blue: West; green: Northwest; magenta: North; cyan: Northeast;
black: Southeast). (b) Maximum VMT-induced WL (mm) detected at each buoy according to color scale at top of
map. White markers indicated that the VMT was not detected at the buoy. (c) Persistence time of the tsunami
signal at each buoy, representing the length of time that each buoy recorded at 1-minute resolution (hr). (d) WL
response to the VMT and marine tsunami at five deep-water buoys in the Pacific using same color scheme as (a).
Two buoys are given on the same line (B52402 and B21420) since their physical locations were on nearly the
same great circle path from Tonga; other buoys are offset 10 cm vertically from each other. VMT arrivals based
on a theoretical travel time of 1115 km/hr[-1] are indicated by grey vertical lines, and marine tsunami arrivals based
on an average travel time of 700 km/hr[-1] are indicated by orange vertical lines. Maps made in MATLAB using
data from Natural Earth.

### 4.3 Coastal characteristics of VMTs

As the VMTs propagated from deep water to the coast, we observed several cases in which an abrupt change in geometry produced a large amplification. We return to the example of buoys 52402 and 21420 discussed above, and now compare data from the buoy closer to Japan (21420) with the nearest coastal tidal gauge that also has $P_A$ data, Kushimoto, Japan (Figure 3). The first Lamb wave with a pressure change of ~0.6 mb occurred at ~1130UT, 15 January at Kushimoto (Fig. 3(a),(c)). The WL response in the $P_B$ record (a positive ~30mm spike then a ~30mm negative one) is direct and presumably represents the forced wave. We compare the two closest $P_A$ records to the $P_B$ data (Aburatsu and Kushimoto; see Appendix A for details). Longwave celerity at the buoy depth of 5700m is 850km hr$^{-1}$; $a_n = \frac{V^2}{V^2-c^2} \sim 2.4$, relative to the observed amplification of $\alpha \cong 4$. The CWT scaleogram in Fig. 3(e) shows the WL response to the shockwave at ~10hr post-eruption as two relatively distinct bands of energy with periods of ~1hr and 5-10min; these fade within ~1.5hr.

Kushimoto WLs effectively illustrate the potential for amplification of VMTs. The first (VMT) waves arrived between 1200 and 1450UT (Figs. 3(b),(d)), prior to the marine tsunami at about 1450UT; their period is ~0.3hr (Fig. 3(f)); shorter-period energy is seen only after the arrival of the marine wave. The initial positive VMT amplitude of ~210mm is a response to the atmospheric shockwave and represents an amplification factor of ~7 relative to the forced wave, and $\beta$~35 relative to the VMT magnitude, for which the inverse barometer response would be only 6mm. Apparently, the Japan trench with depths to 8km ($a_n \approx 5.5$) and continental shelf between buoy 21420 and Kushimoto allowed considerable growth of the forced wave relative to Fig. 3(a),(c). A large volcanic explosion (such as Krakatoa) could yield a shockwave with a magnitude of 30-60mb (Schufelt, 1885), which could potentially drive a large VMT at this location before the arrival of the marine tsunami.

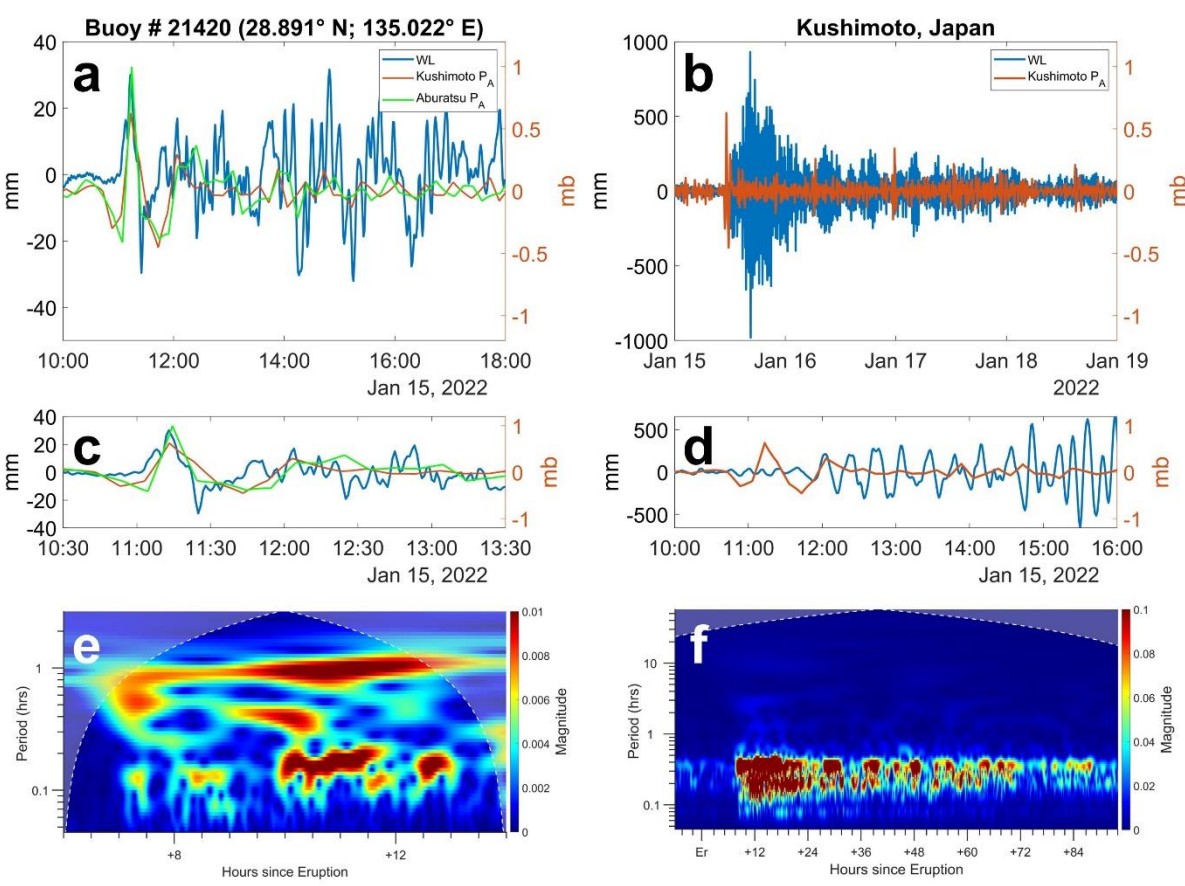

**Figure 3.** Tsunami response at NOAA $P_B$ buoy 21420 and a coastal tide gauge (Kushimoto. Japan):  (a) Residual
$P_A$ at Kushimoto (orange) and Aburatsu (green), and detided residual buoy WL (blue) with $P_A$ records shifted plot
26 and 16min to account for distance from the buoy (see Appendix A for details); (b) $P_A$ (orange) and detided
residual WL (blue) at Kushimoto; (c) expanded view of (a) showing arrival of a VMT as a supercritical forced
wave at 1150UT, ahead of the marine tsunami wave at 1450UT (c); (d) expanded view of (b) showing the arrival
at Kushimoto of the VMT as a subcritical free wave at 1200UT; (e) buoy residual WL CWT scalogram, 6-14hr
post-eruption; (f) Kushimoto WL CWT scalogram for 92hr post-eruption.


Observations near Hilo, Hawai'i show similar phenomena to those observed at Kushimoto (Figure 4).
We use NOAA tsunami bottom-pressure ($P_B$) buoy 51407 in 4.7 km water depth south of Hilo combined with
atmospheric-pressure ($P_A$) and WL data from Hilo (NOAA station 1617760). Fig. 4(a),(c) show $P_A$ and $P_B$ data
(converted to WL). Despite the distance (~100 km) between the two records, the WL and $P_B$ responses are almost
simultaneous, at 0854 UT.  The first $P_A$ pulse of ~1.5mb elicits a WL response of ~30mm ($\alpha$~2) of the same sign,
as expected for a super-critical wave and similar to the response at Kushimoto. This modest amplification is still
slightly larger than expected for $a_n$~1.2. Smaller positive WL pulses follow the first; after the third, these pulses
are overlain by the beginnings of the marine tsunami signal at ~1030 UT. These may be a soliton train, as predicted
by the nonlinear theory (Pelinovsky et al., 2001). The CWT scalogram in Fig. 4(e) shows that marine tsunami
waves with periods of 0.15-0.2hr arrived at buoy 51407 before 1100 UT; shorter waves (periods <0.1hr) arrived
later, confirming the weakly dispersive character of waves in the tsunami band. The VMT is also clearly visible.
It appears just before 0900 as a broadband signal with periods of 0.4-1.1 hr. Over time, the pulse shifts to higher
frequencies and then disappears by ~1200 UT.
The detided residual WL data at Hilo present quite a different appearance from the offshore $P_B$ data (Fig.
4(b),(d)). The first wave arrival (~120mm) occurs at 0928 UT (~1 hr after the $P_A$-spike) with a *negative* excursion
rather than a *positive* one. This is followed by a series of smaller oscillations leading up to the arrival of the marine
tsunami at about 1137 UT. The forced wave is not evident, and the early arriving VMT waves at Hilo are likely
free waves that have propagated around the island on which Hilo sits and then amplified, having been generated
at the abrupt rise of the island platform; the total amplification is $\beta$=9. The waves from the marine tsunami wave
reach ~400mm, which represents an amplification factor of about 5 relative to the same $P_B$ waves at the buoy.
Records from nearby Hawaiian gauges show similar features. The CWT scalogram for Hilo WL in Fig. 4(f)
emphasizes the absence of longer period tsunami waves with periods around 1 hr. Instead, the weak VMT WL
response is followed by waves with similar periods, ~0.15 to 0.7 hrs. Over the next several days, the oscillations
weaken, with the shortest period waves disappearing first. Hilo is well known to be resonant to tsunamis, and our
observations may be related to quarterwave resonance (Pattiaratchi and Wijeratne, 2015; Tang et al., 2017).
However, water levels at other Hawai'ian tide gauges behaved similarly to Hilo.

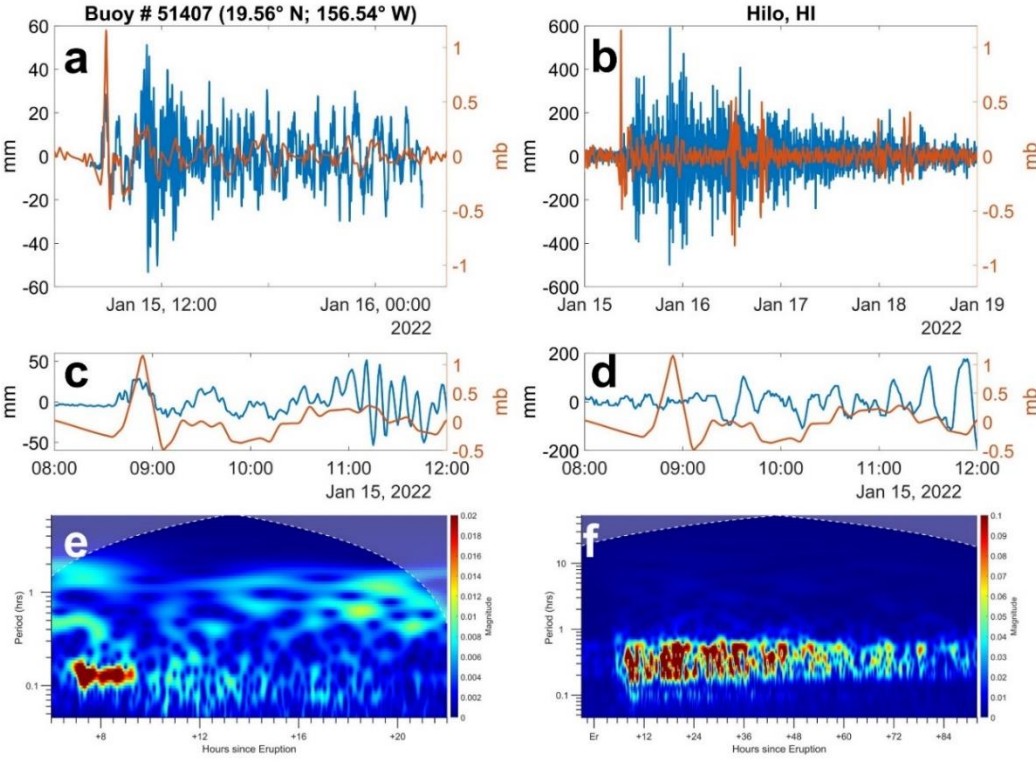

**Figure 4.** Comparison of WL (blue, mm) and $P_A$ (orange, mb) at offshore buoy 51407 and Hilo, HI. (a) $P_A$ at NOAA tide gauge 1617760 Hilo, HI and detided WL residual from NOAA $P_B$ Buoy 51407 south of Hawai'i following the Tonga Event; (b) $P_A$ and detided residual WL (blue) at Hilo; (c) expanded view of (a) showing the arrival of the VMT at Buoy 51407 in the form of a supercritical forced wave at 0854 UT, ahead of the marine tsunami wave arrival at ~1054 UT (c); (d) expanded view of (b) showing the arrival at Hilo of the VMT in the form of a subcritical free wave at 0928 UT; (e) a CWT scalogram of buoy heights from $P_B$ for hr 6-24 post-eruption; (f) a CWT scalogram of WL measured at Hilo for 92hr post-eruption.

Kushimoto and Hilo are only two examples of VMT effects in the Pacific. VMT-induced WL magnitudes

were similar to Kushimoto at other Japanese locations and were 50-210mm in New Zealand and Eastern Australia.

Much smaller (~20mm) VMTs were seen in the South China Sea, though 1-min data were available at only two

locations (Hong Kong and Shenzhen; Wang et al., 2022). In the Eastern Pacific, distant from Tonga, VMT waves

arrived 3.5 (California) to 5hr (Chile) before the marine tsunami, allowing their WL effects to be easily

distinguished (Fig. 1(a),(c), Table S3), and both regions had particularly large maximum tsunami magnitudes

(positive and negative swings). Air pressure ($P_A$) spikes of ±0.6-0.7mb and +1.5 and -0.8mb at Port San Luis, CA,

and Coquimbo, Chile (Figure 5) led to wave excursions of +110 and -150mm, respectively, with total

amplifications of $\beta$~15-25 at Port San Luis (Fig. 5(c), and ~6 (positive wave) and 30-40 (negative wave) at

Coquimbo (Fig. 5(d). There were at least six arrivals of the shockwave over 3d. This recurrence, coupled with

very long decay times (below) caused WL disturbances to continue for >90hr, emphasizing the role of the VMT

in recharging the combined marine and volcanic tsunami (Fig. 5 (e-h)).

These Pacific examples demonstrate combined marine and VMT impacts; in other regions, the VMT

occurs in isolation. At Charlotte Amalie in the Caribbean (Figure 6), the $P_A$-spikes and resulting VMTs are well

correlated (Fig. 6(a)). The first $P_A$-spike of ~1.2mb led to waves of 80mm about an hour later, apparently from

the free wave (Fig. 6(b)). In contrast, the third $P_A$-spike of ~0.5mb apparently excites a forced wave with amplitude

of about 50mm, simultaneous with and of the same sign as the $P_A$-fluctuations (Fig. 6(c)). Waves arriving an hour

later and presumably representing the free wave were larger, ~80mm, giving $\alpha$ =~16. The fourth $P_A$-spike ~±0.2mb

again excited a forced plus free wave response, with the later waves being as large at ±100mm (Fig. 6(d)). This

corresponds to an impressively large $\beta$ = ~30. The CWT scalogram shows that water level in this harbor responds

most strongly at periods of ~0.5 to 0.9hr (Fig. 6(e)). The CWT of $P_A$ at Charlotte Amalie shows eight spikes at

~12hr intervals, suggesting that the shockwave circled the planet at least four times over 4 days (Fig. 6(f); see also

Figure 5g). The largest WL response occurred from the fourth VMT (Fig 6(e), (f)) for yet unknown reasons. Other

gauges in the Caribbean showed significant VMT effects (Figure S11) that were strongest on the second or third

pass of the atmospheric disturbance. While $\beta$ varies with the event, there are numerous volcanoes in the Caribbean,

and severe tsunamis (both VMT and marine) could be a very real hazard in locations where amplification occurs.

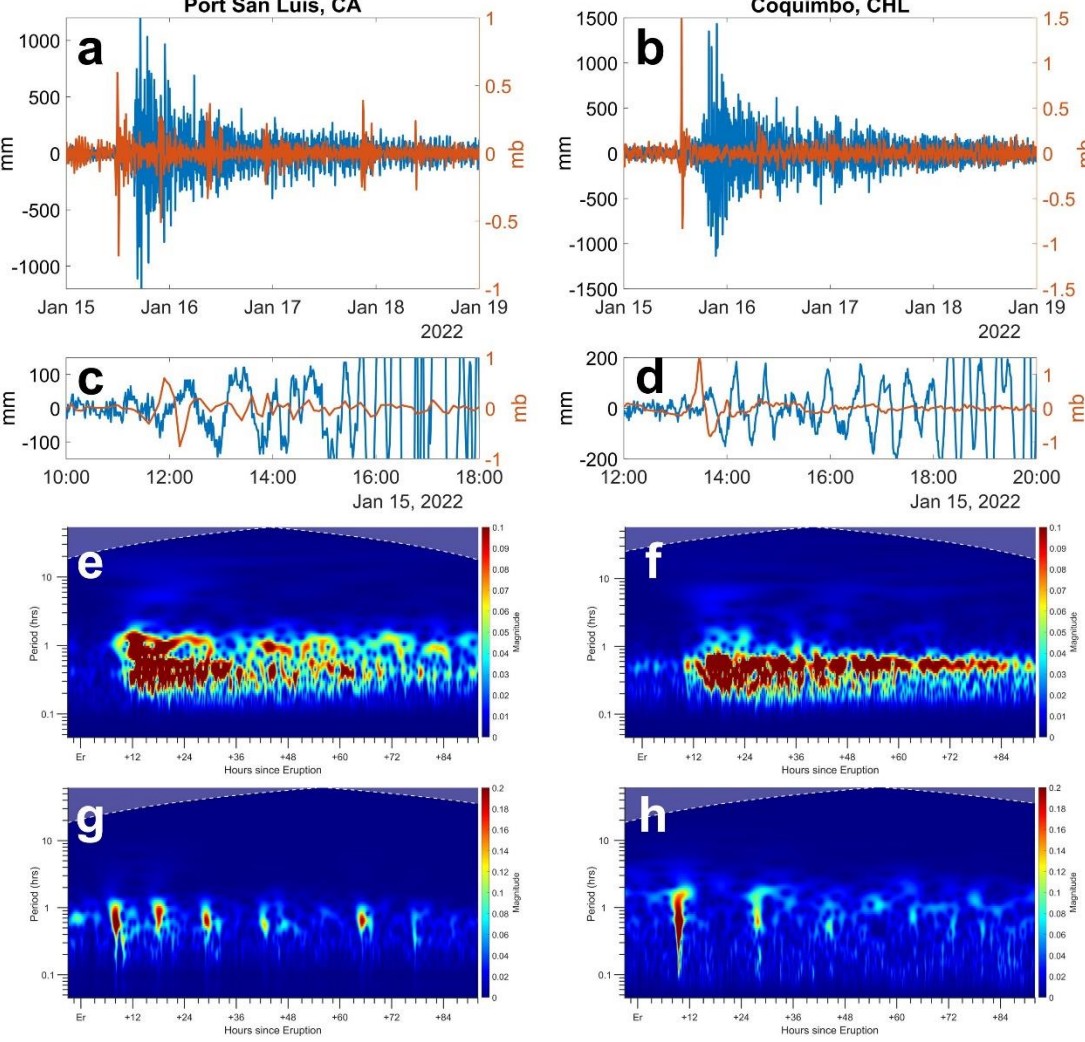

**Figure 5.** Residual WL (blue, mm) and detrended air pressure (orange, mb) at: (a) Port San Luis, California (NOAA Station 9412110) and (b) Coquimbo, Chile; (c) and (d) expanded views of (a) and (b) of the WL and $P_A$ records showing the initial arrivals of the VMT and marine tsunami); and scalograms from CWT analyses of WL in (e) and (f)  and for $P_A$ in (g) and (h). Vertical scales in (c) and (d) are set to a small range to highlight the VMT impacts.

The shockwave magnitudes were generally smaller in the Mediterranean than in the Caribbean, perhaps
because of the greater distance from Tonga and the complex land topography in the region.  Still, VMT-induced
meteotsunamis were measured at many locations; they were largest in Sicily, Sardinia, and the "boot" of Italy.
Because this region is close to the antipode, the first $P_A$ waves arrived from opposite directions only a few hours
apart, at ~2000 and 2330 UTC on 15 January. The propagating waves produced multiple waves rather than a clear
$P_A$-spike that swept across the region. A weaker group of waves occurs 38hr later at ~1200 UTC on 16 January,
followed by a third group at ~0000 UT on 19 January, not seen at all stations. WL records usually show a single,
long-lasting event following the first $P_A$-packet arrival, with muted responses for the second and third packet.  The
largest tsunami amplitude, ~300mm (Figure S12), occurred at Crotone, Italy after a steady build-up from the VMT
arrival. At a small number of stations, e.g., Cagliari, Italy, there were multiple VMTs, as in the Caribbean (Figure
S11). Finally, a few locations in the Adriatic Sea had no response to the first wave packet but responded strongly
to the second VMT, with $\beta \approx 8$ -13.  Thus, the discrete response of WLs to individual shockwaves is not as clear
in the Mediterranean as in the Caribbean, though repeated passes of the shockwave lead to sustained WL
variability.

***4.4. Energy decay***

The Tonga event released significant energy and its tsunami persisted longer in the Pacific than other

recent marine tsunamis. Our estimate of energy $E_o$ for the Tonga Event (0.0096m$^2$, N= 37) is comparable to the
Chilean event (0.01m$^2$; N =28) and about 3.8x less than the Tohoku Event (0.036m$^2$; N=40). Previous estimates
for the Chilean and Tohoku Events were 0.009 m$^2$ and 0.032m$^2$, respectively (Rabinovich et al., 2013). Decay
time scales for the Tonga Event varied from 29-44hr in Alaska, 25.4hr (Santa Barbara) to 37hr (San Diego) on
the US West Coast, and 22.2hr (Nawiliwili, Hawaii) to 29.3hr (Pago Pago, Samoa) for island stations (Figure
S19). The Tonga decays are notably longer than other events, especially in Alaska and (most) California locations.
The differing timescales between stations depend on distance from the event, frequency content (high frequency
decays more quickly), and shallow water processes (Rabinovich et al., 2013). Our estimated median $t_d$ values for
the Tohoku, Chile and Tonga events are 26.6±2.4hr (N=40), 27.6±2.8hr (N=27) and 31.0±2.6hr (N=37),
respectively (Figure 7). Previous estimates for the Tohoku and Chilean Events were 24.6 and 24.7hr. The longer
decay time of the Tonga Event emphasizes the importance of the VMT. Though the VMT was smaller than the
marine tsunami, it was refreshed by the Lamb waves that repeatedly circled the planet (see e.g., Figure 5g). The
long energy decay scales calculated for the Northern Pacific are in line with our simple estimates of decay taken
from the buoys; which were longest in the Northern/Northeast Pacific and near Tonga (e.g., Hawai'i and Pago
Pago; see section 4.2).

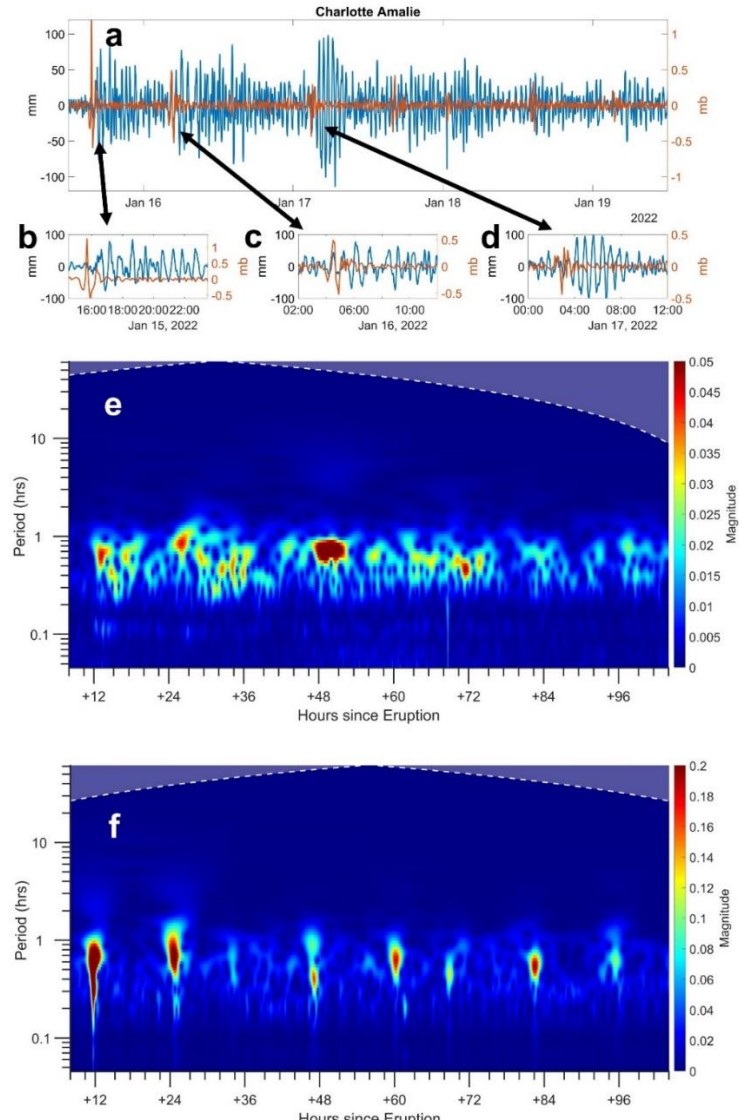


**Figure 6.** VMTs at Charlotte Amelie (NOAA gauge 9751639) in the Caribbean: (a) Residual WL variability (blue) and $P_A$ (orange) from UT 15 to 19 January 2022; (b)-(d) expanded views of (a) at the times of the 1st, 2nd, and 4th $P_A$-spikes; (e) and (f) CWT scalograms of the WL and $P_A$ records in (a).

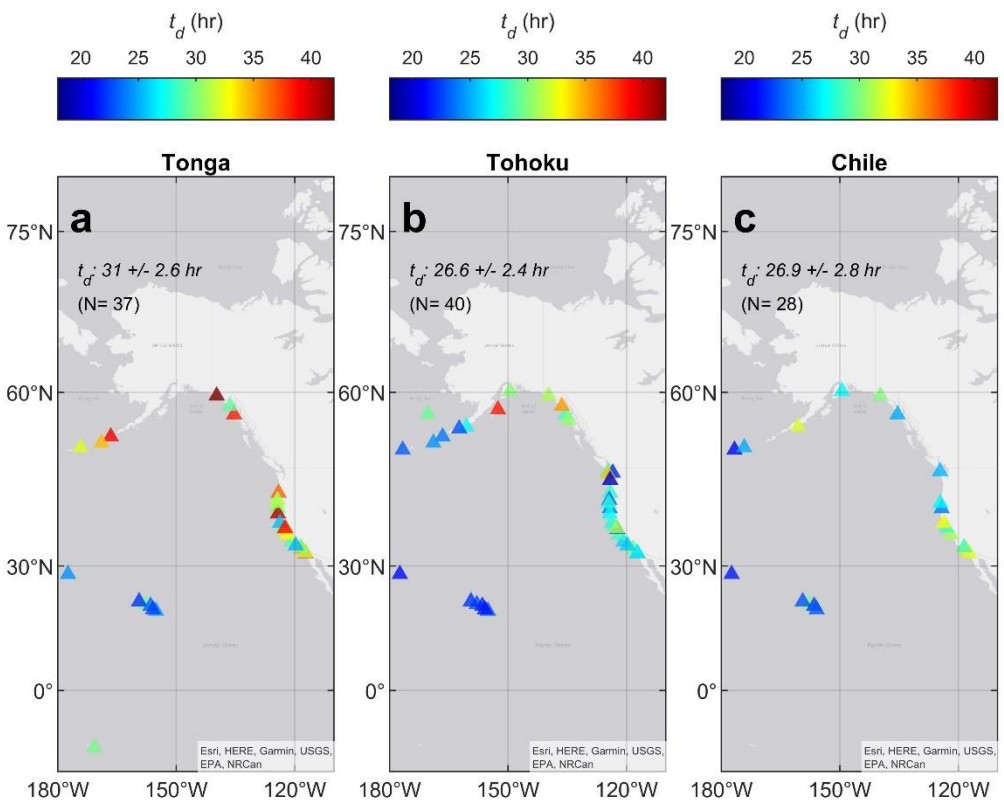


**Figure 7.** Decay timescales (hours) of recent tsunami events at NOAA gauges in the Northern Pacific; showing (a) Tonga; (b) Tohoku; and (c) Chile. Median $t_d$, errors, and number of stations used are given in each panel.

### 4.5. Amplification, β

Amplification $\beta$ is a vital indicator of possible future VMT hazards and vulnerability. It was calculated for ~75% of all tide gauge locations where the shockwave was detected in a nearby $P_A$ record (Tables S5 and S6). Clearly, $\beta$ is highly local, with strong spatial heterogeneity (e.g. Figure 8). Maximum values of 15-35 were measured at 26 stations in all regions, and over 50 locations had $\beta >10$ (Figure 8 (a-d)). The largest values of $\beta$ are observed in Japan, the Northeast Pacific, New Zealand and Australia, and the Caribbean. Wherever high $\beta$ values are observed near an active volcano, there is the potential for a large VMT. Note that $\beta$ values are uncertain

by ~30% (see Appendix A), mainly due to the uncertainty of $P_A$ observations which have low amplitudes and
coarse temporal resolution.

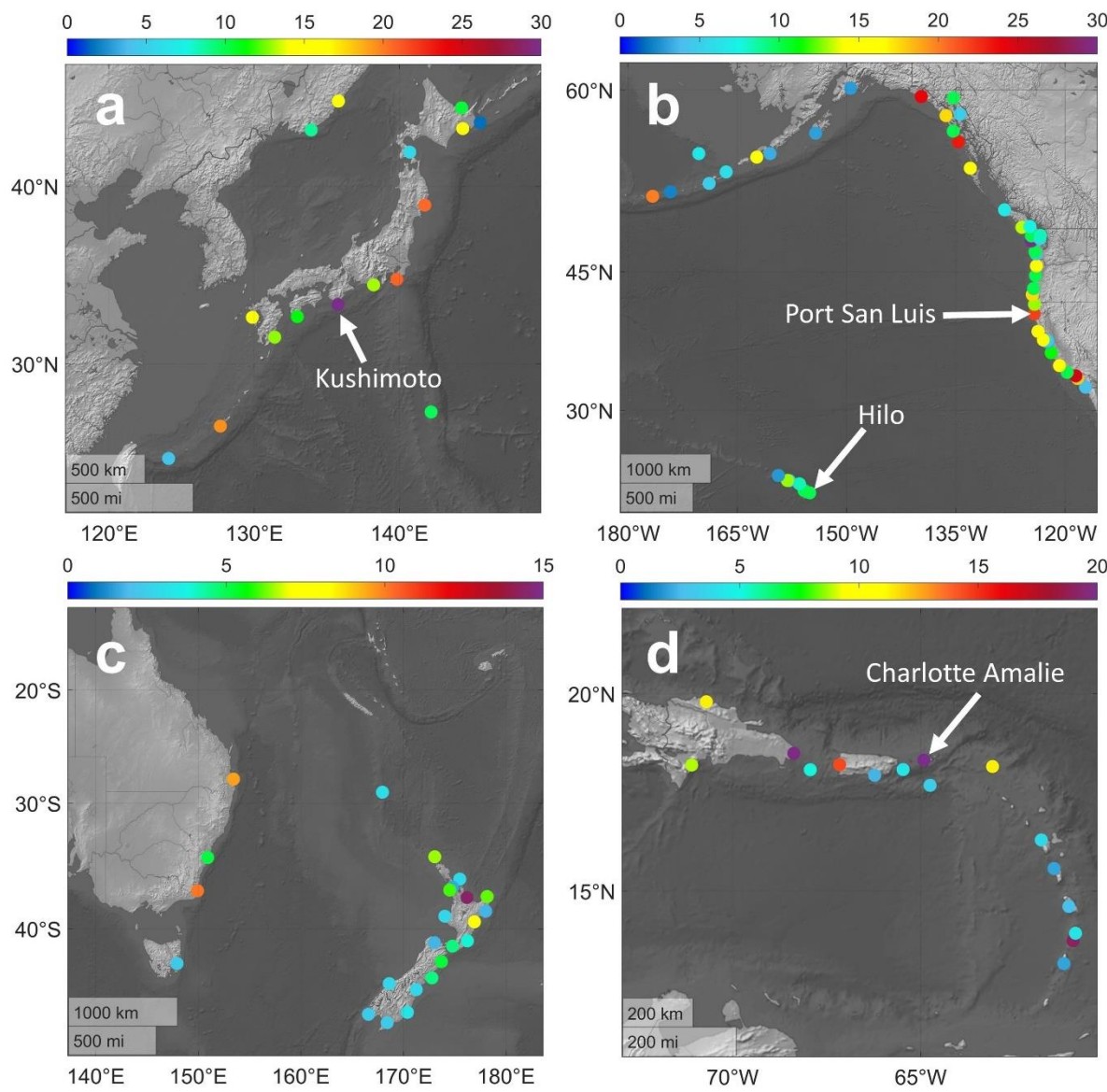


**Figure 8.** Amplification, $\beta$ in: (a) Japan; (b) the Northeast Pacific; (c) New Zealand and Australia; and (d) the Caribbean. Locations of note with large amplification which were discussed above are indicated; Kushimoto (Fig. 3), Hilo (Fig. 4), Port San Luis (Fig.5), and Charlotte Amalie (Fig. 6). Note diverse color scales. Maps made in MATLAB using data from Natural Earth.

## 5. Discussion

Analyses of high-resolution WL data from tide gauges (with local $P_A$, where possible) provides an unprecedented global view of a volcanic meteotsunami (VMT) acting together with a marine tsunami. A moderate marine tsunami was measurable at nearly all Pacific Ocean tide gauges and deep-water buoys, but at only a few stations elsewhere. In addition, most tide gauges and about half of the deep-water buoys also observed the VMT. In the North Pacific, wave amplitudes and energy were comparable to the Chilean Event. Out of 308 tide gauges, 10 showed a total VMT amplification ($\beta$) of > 20, 54 were >10, 113 were >5, 204 were 2 or more, and 230 were 2 or less; the remainder did not register any detectable VMT signal. Hence, significant amplification is a localized, but still potentially important, process. We note that much of the world's coastline is still not gauged, and there are many locations in which the VMT was amplified, but not measured, e.g., Lima. Thus, the Tonga Event tsunami was "global" because of the reach of the VMT and its impacts on WLs.

In the Pacific, the VMT preceded the marine tsunamis by up to five hours and the two together produced observable perturbations in water levels for more than three days after the eruption. The effects of atmospheric gravity waves were observed in ocean bottom pressure data after the arrival of the Lamb waves and before the marine tsunami arrived. However, we observed a delay (~1-2 hours) of the water level response to the shockwave at coastal tide gauges. This delay may be related to the "sequencing" of tsunami waves and observations that the first wave of a tsunami is not always the largest (Okal and Synlokas, 2016). However, this suggestion is based on "traditional" seismic tsunamis; it is not clear if VMTs follow exactly the same physical dynamics.

How can we place the Tonga event in a larger context? This event drove VMTs no larger than ~210mm in the far field due to shockwave magnitudes of ~0.5 to 5 mb. However, the total amplification, $\beta$, varied from ~1 to 35×. Values at the larger end of this range were mainly seen at coastal locations; island locations typically had $\beta$ <5, with only a few exceptions (e.g., Hawai'i and Naha). The reasons why certain regions exhibited a larger amplification (e.g., $\beta$) than others, and the possible role of bathymetry, remain to be understood, e.g., through model studies like Denamiel et al. (2022). It seems likely, however, that locations with an ocean trench between the source and the coastal station are at particular risk; this is typical for much of the Pacific "Ring-of-Fire", as conceptualized in Figure 9. We assume a 5mb Lamb wave travelling over deep water which initially induces a forced wave WL fluctuation of 60mm. After travelling some distance, the forced wave grows four-fold. The trench, with $F_a$ near unity, increases VMT amplitude even if the trench is narrow relative to the wavelength of the longer-period tsunami components. Coastal and harbor processes, which can vary substantially along a coast, provide a further boost. Taken together, these processes can an amplification of up to $\beta = 36$ (as suggested in

Figure 9), in which case an initially modest (5mb) $P_A$-spike and corresponding WL fluctuation of 6cm can become
a ~1.8m tsunami.

The VMT from the Tonga event was small, but $\beta$ was >10 in many parts of the world with active

volcanoes, including Italy, Alaska, Japan, and New Zealand.  A much larger VMT can occur close to a VEI 6-7
volcanic explosion. For example, in 1883, ship barometers measured fluctuations of 1-2 inches of mercury (30-
60mb) near Krakatoa (Symons, 1888). Taking 30mb as a conservative upper limit for a VEI 6 event and $\beta$ = 10 to
35, a VMT of 3.5 to ~10m is possible. In most cases, this would be later followed by larger water waves, but the
rapid arrival of VMT waves of this size could be catastrophic and might occur in some locations without being
followed by a marine tsunami. Moreover, Krakatoa is not the largest historical event by any means—the Santorini
(~3600YBP) and Tambora (1817) events were much larger (Newhall and Self, 1982), but these events lack data
regarding VMT impacts.

Present-day warning systems are designed for marine tsunamis, and do not generate timely warnings for

meteotsunamis of any origin, as noted by Vilibić et al. (2016). Future warning systems should consider both
marine and meteotsunamis, but this is not straightforward, because of differences in the causation and warning
times between weather and volcanic meteotsunamis. Weather conditions for meteotsunami genesis, which evolve
over days, may be able to be at least partially predicted, and this threat is confined to specific regions. Volcanic
eruptions are a different problem.. The VMT threat is global, VMTs can cross an ocean basin in a matter of hours,
given the rapid shock wave celerity (~1100 km hr$^{-1}$), and their magnitude can be larger.  Thus, though VMTs
occur only infrequently, the possible hazard  deserves further consideration.

### 6. Conclusions

We conclude the following regarding the volcanic meteotsunami (VMT) from the Tonga Event:

•   The VMT arrived before the marine tsunami at all stations where both were observed, though the

marine wave was larger at stations where both occurred.

•   The atmospheric shockwave transited the globe multiple times; on every pass it imparted additional

energy to WL fluctuations which sustained or re-excited the VMT, likely contributing to a ~25%

longer decay timescale than for recent marine tsunamis generated by earthquakes.

•   The re-focusing of the shockwave in the atmosphere near the antipode of the eruption may have

increased tsunami amplitudes in Africa and the Mediterranean. The reasons for the strong Caribbean

response are yet unclear.

• The first wave observed at deep-water pressure gauges was the super-critical VMT-forced wave
predicted by theory, but at most tide gauges only the sub-critical free wave response was observed.
• The nominal amplification, $a_n$, shows that deep water allows strong growth of the forced wave
beneath a VMT (Proudman resonance). The large total amplification, $\beta$, at Japanese coastal stations
suggest that deep water trenches around the Pacific "Ring-of-Fire" (with its many volcanoes) and
elsewhere may produce the potential for large, consequential VMTs.


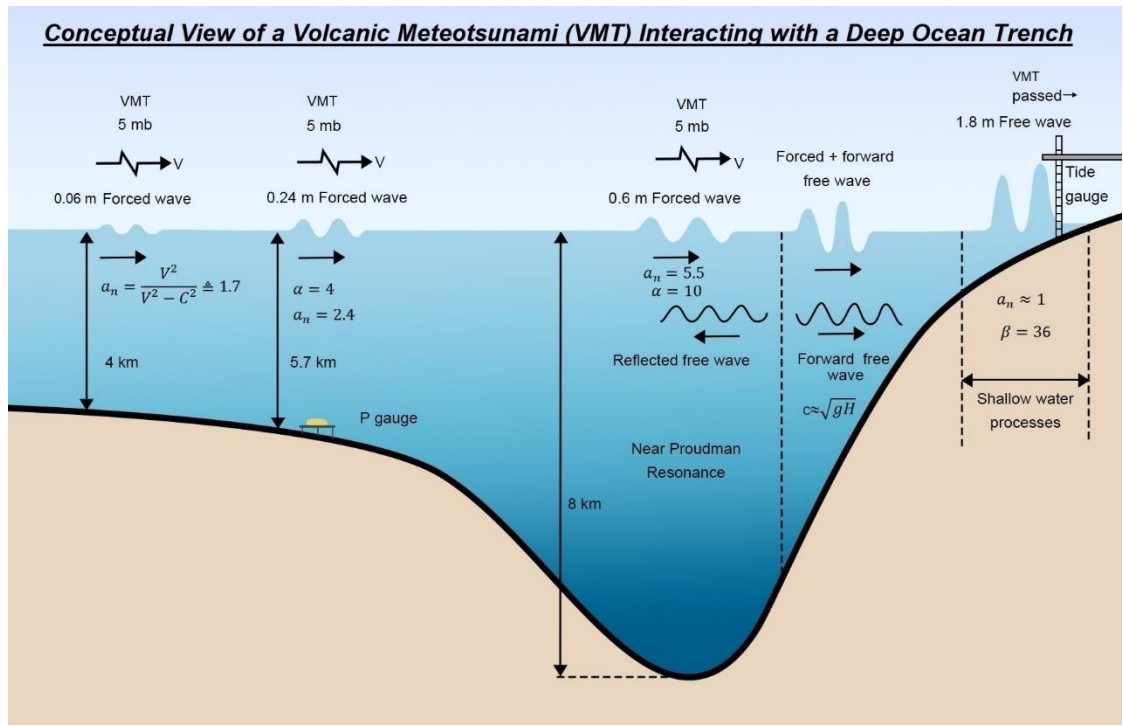

**Figure 9.** Conceptual view of amplification of a VMT, based on Tonga-Event observations. An initial shockwave
amplitude of 5mb is amplified by Proudman resonance in the trench, and again by shallow water processes, after
reflection of a free wave by the steep topography landward of the trench. With $\beta = 36$, a 1.8m tsunami occurs at
the tide gauge. A larger VMT would lead to a proportionally larger response.

**Appendix A: Extended Details of Materials and Methods**

*A1. Data Inventory*

We acquired one-minute resolution data from the following sources: the European Commission (EC) World Sea Levels Database (https://webcritech.jrc.ec.europa.eu/SeaLevelsDb/Home), the Intergovernmental Oceanographic Commission (IOC) sea level station monitoring facility (https://www.ioc-sealevelmonitoring.org/; VLIZ, 2022), the National Oceanic and Atmospheric Administration (NOAA) CO-OPS Tides and Currents tsunami warning network (https://tidesandcurrents.noaa.gov/tsunami/), and Land Information New Zealand (https://www.linz.govt.nz/sea/tides/sea-level-data/sea-level-data-downloads), plus data obtained by direct communication from the National Institute of Water and Atmospheric Research (NIWA) of New Zealand (https://niwa.co.nz/our-services/online-services/sea-levels). Other stations from these networks with less frequent data were used when 1-min data were not available. Tidal predictions and residuals are provided in the EC and NOAA databases, however, a tidal signature or a slope sometimes remains in the provided residuals, and the IOC and NIWA data does not provide any predictions. Therefore, we apply an EEMD analysis (Huang et al., 1998) to all WL data to remove low frequency components and biases in mean water level to yield data in which the tsunami signal is dominant.

Air pressure ($P_A$) records at 1-minute resolution is downloaded from the Chilean Meteorological Directorate (CMD; https://climatologia.meteochile.gob.cl/), the Australia Bureau of Meteorology (BOM; http://www.bom.gov.au/climate/data/), and the Instituto Superiore per la Protezione e la Ricerca Ambientale (ISPRA; https://www.mareografico.it/) network for Mediterranean locations, 6min $P_A$ data is downloaded from NOAA at tide gauges and $P_B$ data from offshore buoys in the Pacific and Caribbean (https://tidesandcurrents.noaa.gov/stations.html?type=Meteorological+Observations; https://www.ndbc.noaa.gov/obs.shtml), and 10-min $P_A$ data is acquired from the Japan Meteorological Agency (JMA; https://www.data.jma.go.jp/obd/stats/etrn/index.php) and the National Institute of Water and Atmospheric Research National Climate Database (NIWA/NCD; https://cliflo.niwa.co.nz/). A total of 137 air pressure locations were used, listed in Table S5.

Finally, we download data from 30 Pacific deep-water buoys (see Table S1) from the National Data Buoy Center (NDBC; https://www.ndbc.noaa.gov/obs.shtml) tsunami warning center operated by NOAA; these provide 1-min data during "active" WL events and 15-min data otherwise. Other buoys were investigated, but because the

buoys only sometimes operated at 1-min resolution, many were not triggered until the VMT wave had passed;
thus, it was most often not captured. All buoy data and air pressure data were conditioned using EEMD as
described above.

*A2. Water Level (WL) Analysis*

VMT magnitudes and arrival times, and the amplitudes of the largest positive and negative tsunami

waves at each location are determined from the WL residuals via numerical and visual estimation of the residual
time series. The "first arrival" times and amplitudes represent the effects of the VMT, which travels faster than
the marine tsunami; times are determined by finding the rising edge of the first obvious anomalous wave in the
residual WL time series, and the VMT amplitude is defined as the maximum WL immediately after the first arrival
(Table S3). At a small number of locations, the VMT wave could not be clearly observed, as noted in Table S3,
and in Figs. 1(c),(d). We compare the distances and first arrival times at all tide gauges stations via robust
regression (Holland and Welsch, 1977) and find an estimate of the VMT velocity from the slope of the regression
as $1054\pm7$km hr$^{-1}$ (Figure S11(b)), slightly less than that estimated from the air pressure gauges ($1115\pm3$ km hr$^{-1}$;
Figure S11(a)). These estimates can be compared to the much slower celerity estimate for the water wave
component of the tsunami ($708\pm8$ km hr$^{-1}$; Figure S11(c)), clearly demonstrating that the "first arrival" WLs are
due to the VMT. Note that the water-wave celerity corresponds to an average water depth of about 5km.

The timings and amplitudes of the largest positive (negative) waves due to the marine tsunami are

determined by when the first local maximum (minimum) occurs after the first arrival of the tsunami. At some
locations, slightly larger amplitudes are seen many hours later, usually on the following tidal cycle (i.e., "tidal
pulsing"), while other locations have the largest wave arriving a few oscillations after the arrival; the latter may
be due to the issue of "sequencing" as described by Okal and Synolakis (2016). WLs and times for maximum
WLs, as well as the differences between extreme levels and the VMT arrival are given in Table S2 and Fig. 1(a),(c)
and Figure S5 and S6, and the same parameters for minimum WL are provided in Table S4 and Figures S7 and
S8. The time differences between "first arrival" and max/min WLs are shown in Figures S9 and S10.
Determination of VMT ("$P_A$-spike") amplitudes was carried out in the same manner as for the tsunami amplitudes.

*A3. Air-pressure gauge choices for Kushimoto*
Comparison of the Kushimoto tide gauge WLs to offshore buoy #21420 and air pressure (Figure 3) raises
the difficulty that there is no $P_A$ station within more than 300km of the buoy; we use, therefore, the two nearest.
Aburatsu (~465 km) is on a direct line from Tonga and the buoy, while Kushimoto is 305 km from the circle
centered on Tonga through the buoy. Accounting for the distance between the coastal gauges and the buoy using
a shockwave velocity of 1092 km hr$^{-1}$ (Table S3), we shift the time index of the $P_A$ records by 16 and 26 minutes,
respectively. Both $P_A$ records are used, because the sparse, 10 min, resolution of the $P_A$ records precludes either
from completely capturing the VMT.
*A4. Energy Decay Analysis*
Following (Rabinovich, 1997), we detide 1-min NOAA WL data, remove any residual trend, and then
produce power spectra for 4hr segments of the WL residual, with an overlap of 2 hours between successive
analyses. A multi-tapered method (McCoy et al., 1998) was applied, because it reduces noise and edge effects,
but still conserves energy.  The energy within the tsunami band (between 10 minutes and 3 hours) was then
integrated for each 6hr period and an exponential decay model of form $E = E_o e^{\frac{-t}{t_d}}$ applied, where $E_o$ is the peak
energy in the fit and $t_d$ is the e-folding (decay) time scale.  To account for the initial "diffusion period" (Van Dorn,
1984; 1987), the two initial, largest energy values were removed; hence, $E_o$ represents the energy at the
commencement of exponential decay. The exponential decay was fit to all tsunami-band energy values until
measurements dipped below the noise floor.  The noise floor was defined as the 80% percentile energy in the
tsunami band from 7-12 days after the event.  Each fit was examined for validity, and the range of points in the
fit was manually adjusted in five cases.  For fits for which the standard error in the coefficients was more than
20%, the coefficient value was removed.  The analysis was applied to four events:  The 2009 Samoa tsunami, the
2010 Chilean tsunami, the 2011 Tohoku tsunami, and the 2022 Tonga tsunami.  However, due to the low energy
of the Samoa event, we focus primarily on the other three.  In our analyses, we also distinguish between coastal
and island stations.  Unfortunately, high resolution DART data are not presently available over a sufficiently long
time scale to repeat the analysis of (Rabinovich et al., 2013) exactly.
*A5. Uncertainty and Errors*
The possible sources of uncertainty in this study arise from:
1) Instrumental accuracy: Measurements of WL at most locations considered report values to an accuracy of 1mm,
and US locations from the NOAA tsunami network are only reported to an accuracy of 10mm. Values are reported
to this accuracy in figures and tables. However, due to oceanographic noise from coastal waves and other
processes, a "noise floor" of at least 10 mm is likely at all locations. Thus, we assume all locations have an
uncertainty of ±10mm in the calculations of $\beta$ below. This noise level represents a small uncertainty in the
determination of maximum and minimum tsunami heights, e.g., a 1000mm tsunami wave would have a relative
error of 1%. However, there will be a larger relative error in the estimation of the VMT WL amplitude, e.g., a 20-
200mm VMT WL would have a relative error of 5 to 50%. All $P_A$ readings are reported to an accuracy of 0.1mb.
Since the $P_A$ fluctuations are mainly in a range of 0.5 to 2.0mb, the instrumental error may be up to 20%.
2) Mean offset/bias in residuals: Common estimates for tidal prediction, such as those performed in the
downloaded residual products here, subtract tidal components from water levels using harmonic analysis methods,
which are typically based on past epochs and may not always remove all tide-related fluctuations or may include
a bias due to sea-level rise or other oceanographic processes (Jay, 2009; Zaron and Jay, 2014; Devlin et al., 2014;
Devlin et al., 2017; Devlin et al., 2021; Fang et al., 1999). These artifacts may give erroneous estimates of tsunami-
related WLs. Our application of EEMD to further separate and remove leftover tidal components in the lower
modes of the decomposition largely alleviates this issue. Analyses of the mean values of residuals WLs after the
EEMD conditioning show that almost all residual time series have a mean value $\ll$ 10mm, a problem no larger
than the instrumental accuracy issue. However, we still subtract the mean bias from our reported results of WL
(max/min tsunami waves and VMT amplitudes). Similarly, the EEMD process also removes diurnal and low-
frequency variability in $P_A$, and analyses of the residuals show that all locations have mean values less than
0.001mb. Thus, the offset or bias in $P_A$ values is insignificant in relation to the instrumental accuracy.
3) Coarse temporal resolution: Nearly all WL data used here are 1-min resolution. This is sufficient in the
estimation of the marine tsunami and VMT related waves, which have frequencies of ~5 min to a few hours.
However, only some of our $P_A$ data is at 1-min resolution (Italy, Chile, and Australia), the remainder is 6-min
resolution (US) or 10-min resolution (NZ and Japan). The pressure wave is a rapidly changing phenomenon which
shifts from strongly positive to strongly negative over a short time (20-60 min) Therefore, it is possible that the
$P_A$ spikes may not be fully captured in the coarser resolution data and may misrepresent the actual intensity of the
VMT wave. This unavoidable problem is the largest source of uncertainty in our study. We account for this by
qualitatively increasing the uncertainty values of the instrumental accuracy for $P_A$ (±0.1mb) to ±0.15mb for the 6-
min data and ±0.2mb for the 10-min data.
The calculation of $\beta$ divides the VMT WL by the $P_A$ spike; i.e., $\beta = \frac{WL_{airshock}}{P_A}$ . We determine the relative
error in $\beta$ by propagating the uncertainties detailed above as: $\frac{\delta\beta}{\beta} = \sqrt{\left(\frac{\delta WL}{WL}\right)^2 + \left(\frac{\delta P_A}{P_A}\right)^2}$; $\delta WL$ is 10  mm, $\delta P_A$ is
0.1mb at 1-min stations, 0.15mb at 6-min stations, and 0.2mb at 10-min stations. Using these error estimates, 21
locations have relative uncertainties in $\beta$ which are greater than 50%, four of which are greater than 100%
(statistically insignificant). The overall average uncertainty is 30.8%.  Best results were found for 1-min pressure
data (e.g., Chile had an average of 16% and Australia had an average of 13%), and somewhat less accurate results
for 10-min pressure data (e.g., Japan and New Zealand both have averages of 27%). However, the largest
uncertainties occurred in places where VMT amplitudes were very small, regardless of air pressure data resolution.

**Code and Data Availability** All data used in this study are deposited in an online repository of the Harvard
Dataverse at: https://doi.org/10.7910/DVN/F0G63H. Datasets included are original 1-min water levels, post-
EEMD water level residuals, original air pressure data (1-minute, 6-minute, and 10-minute resolution), and post-
EEMD air pressure residuals. All code was performed in MATLAB and can be shared via direct communication
with the authors.

**Author Contributions:** All authors contributed to conceptualization, validation, visualization, and
reviewing/editing. A.T.D. contributed data curation, formal analysis, investigation, methodology, software, and
text writing. D.A.J. contributed formal analysis, investigation, methodology, and writing and editing. S.A.T.
contributed formal analysis, investigation, software, methodology, writing, and editing. J.P contributed funding
acquisition, supervision, and editing.

**Competing Interest Statement:** The authors have no competing interests.

**Acknowledgments** The authors wish to thank Wong Wai-chung (Alvin) for his helpful discussion and insights
on submarine volcanic eruptions.

**Funding was provided by:**
National R&D Program of China grant# 2021YFB3900400 (ATD, JP)
The General Research Fund of Hong Kong Research Grants Council (RGC), grant# CUHK14303818 (ATD, JP)
Jiangxi Normal University Start-up Fund (ATD, JP)
National Science Foundation grant# 2013280 (SAT)

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
