# Peer review of "Global water level variability observed after the Hunga Tonga-Hunga Ha'apai volcanic tsunami of 2022"

_EGUsphere, 2022_

## Referee Comment (RC2)

This is yet another article about the atmospheric waves and subsequent ocean waves observed after the explosive eruption of the Hunga Tonga-Hunga Ha'apai volcano. Overall, it is an interesting study with detailed analyses of all available observations (i.e., following the open science policies) during and after the volcano explosion. However, the authors tend to oversell their results: grand statements (e.g., title), movies not providing any information used in the article, lack of reference to other studies related to their results, renaming physical processes already fully documented in the literature (e.g., Lamb waves), etc. Consequently, I recommend major revisions following my comments below.

Major comments:

1.  I appreciate the efforts made by the authors to find names for physical processes not named before. However, I have several objections to the way this naming was done in the article. First, Lamb waves are already defined and documented and their name should be kept. The authors cannot single handily decide to rename this process named after Lamb research in 1911. Second, "air-shock" seems to me a poor choice as directly invoking (at least to me) "air-shock absorbers" and not any physical process. I propose "sonic boom" related atmospheric waves (which is based on the physical process occurring and heard during the explosion). I am not imposing this name as I am sure some objections can be easily voiced against it. However, I would like to engage the authors to rethink more deeply of the names they give. Third, these "sonic boom" related atmospheric waves (or whatever other name the authors might chose) include, at least, both Lamb waves (barotropic process) and Perkaris waves (3D internal waves) and maybe others not known to me. Fourth, once a name is decided, the authors must stick with it. Is it "air shock", "air-shock", or "atmospheric" tsunami? Finally, there is already a name for tsunamis driven by atmospheric forcing: "Meteotsunamis". So why using another name? I suppose one argument is that it is not a "Meteo" event … But, in this case, should the "meteorological ground stations" be also renamed as they measured the mean-sea level pressure from the Lamb and Perkaris waves? I propose to keep meteotsunamis with two categories: "weather" and "sonic boom" related events. This approach keeps the historical naming used for more than 20 years in the scientific community and distinguishes between the atmospheric sources (as generally done in the tsunami community that distinguishes the tsunami sources: earthquake, volcano, landslide, asteroid, etc.). More on the naming, why using "Tonga-Hunga-Ha'apai" when the entire scientific community as well as the site the authors refer to (https://volcano.si.edu; line 54 of the article) name the volcano: Hunga Tonga-Hunga Ha'apai. I have no problem to simplify the name to "Tonga event" (and not Tonga-Event; line 483) in the rest of the article but the first mention of the volcano should correspond to the accepted name (to my understanding, the full provided name is not controversial).

2.  Title: I personally do not see the "lessons" learnt from this event in this specific article. The fact that warning systems should include "sonic boom" related (or whatever name you wish to give them) meteotsunamis is already published (https://doi.org/10.1175/BAMS-D-22-0164.1; https://doi.org/10.1002/essoar.10511565.1). Interestingly, the inference of the authors (made with the analysis of the Tonga event observations) that such meteotsunamis (free waves + force waves) can reach 10 m has been demonstrated with numerical models. This leads to my third point.

3. The authors fail to cite a vast majority of the already published literature. I understand that their article is not a review of all the work already done but some "important" references (in the sense of what was already found similar/different to their study) seemed to have be ignored. I encourage the authors to revise their literature review as something cannot be presented as a new result/conclusion if it already has been published. For example, I think the authors should get familiar with the work of Okal and Synolakys (2016) which is nicely related to their free wave analysis and might (or not, I encourage the authors to look at it in depth) be related to the behavior in the Caribbean. Amplification of the meteotsunami waves nearshore depending on the "amplification factor" of given harbors or bays is also a well-known and well-documented characteristic (please review the meteotsunami literature), this might be another reason for the Caribbean response (again to be proven or rejected by the authors).

4. Theories and methodologies presented in the article are what I would call "classic" and, in my opinion, the article tries too hard to sell facts that are already partially known from other studies (on the Tonga event itself but also from the tsunami and meteotsunami communities). Putting together all the pieces of a puzzle is good enough achievement for a scientific article, no need to amplify the novelty and the reach of the results. For example, (1) the additional movies do not add any value to the article, (2) the lengthy description of the Garret (1976) theory does not particularly add to the analyses (reference and brief summary should be enough), (3) the title is "sexy" but not really related to the true findings of the study, etc. However, last figure summarizing all the ocean processes is, for me, the main achievement of the article (finalized puzzle) and could be even improved by better representing the synchronization of the events (e.g., free wave occurs after the forced wave) and being more specific with the shallow water processes (e.g., shoaling, harbor resonance, etc.) which play a really significant role as documented for weather related meteotsunami events.

---

## Author Response (AR1)

Tsunami paper reviewer responses

RESPONSES TO EDITOR:

*Dear Editor,*

*We are here providing a revision of our manuscript (#egusphere-2022-925) submitted to Ocean Science, which is now titled: "Global water level variability observed after the Hunga Tonga-Hunga Ha'apai volcanic tsunami of 2022". We have done our best to address the comments of both reviewers, and to that end have made some substantial changes and omissions. We have shortened our introduction, split our Discussion section into Discussions and Conclusions, and changed some figures. One of the largest global changes to our paper is a modification to our primary phenomenon of investigation, the water level variability induced by the shockwave after the volcanic eruption. We originally named this an "air shock tsunami", but after considering comments and suggestions of the second reviewer, we have decided to call this a "volcanic meteotsunami". Another major change in this version is that we have removed our animations, also based on reviewer suggestions. These videos were uploaded and hyperlinked in the original preprint version of our paper but will not need to be mentioned or linked in the revised version.*

*We also wish to thank the editors for their patience when we needed extra extensions, and their help in facilitating our extra required work.*

*We provide our responses to all reviewer comments below. We hope that our revision will be acceptable in its current form. Please contact us if any more information is required.*

*-Adam T. Devlin (first author)*

**Reviewer #1 Comments:**

**Tha manuscript attempts to quantify Lamb wave-driven global tsunami waves coming out of the Tonga 2022 explosive eruption event. The presented material and methods looks fine to me, I don't see any major flaws there.**

*Thank you for your time and effort in reviewing our manuscript. We will do our best to address your concerns below.*

**However, the methodology is somewhat simplistic and known to some extent, to which detailed repetition of the theoretical background (in Section 2) may help, but may be partially unnecessary (I have no strong opinion on that).**

*We understand that some of our theory in the paper has been presented previously, especially in the meteotsunami literature. However, our reason for including the basic theory is to provide a foundation for the definition of one of our important calculations, the "amplification factor", β. We believe this parameter is a novel part of our study, and that including parts of the fundamental theory provides appropriate context for our amplification factor definition, particularly for those not familiar with the meteotsunami literature. The amplification factor is also the main focus of our final figure. Without a suitable treatment of the foundational theory, the meaning of our amplification factor would be obscured. However, we have carefully looked at this part of the paper and tried to tighten up the text to only provide the most important background.*

**What I can see as a major problem of the manuscript is that it is not discussing already published papers with the same topic - there are several tens of papers on the Tonga 2022 event accessible on the net, yet only 2-3 are mentioned here. Some of them are discussing the topics presented in this manuscript – like https://doi.org/10.1038/s41586-022-05170-6, https://doi.org/10.1007/s00024-022-03154-1, https://doi.org/10.1029/2022GL098752, and some others - so, I suggest to add in the introduction (maybe a separate paragraph) the most relevant papers on the Tonga 2022 event, while discussing your findings with respect to their results in Section 5, including a list of novel results and conclusions that are coming from your study.**

*Thank you for bringing these publications to our attention. It is true that this event has attracted a lot of attention, and there are indeed many papers on this event. Even in the time since we submitted our initial submission to Ocean Science (Sept of 2022), there have been many new publications. It has thus been quite difficult to "keep up" with all these papers. We have now added the papers you have mentioned, as well as many others we have found since receiving this review. Most of the new references are included in the Introduction section (which has been heavily restructured) in the introduction. We have also added more background on weather-related meteotsunamis in Section 2 about the physics of atmospheric tsunamis and made some contrasts between the two types of meteotsunamis. Finally, we have included some of these works in our Discussion and compared our observations and results to those of other researchers.*

**I am aware that the authors might be sligtly annoyed - like I am - with a reaction of a substantial part of the geosciences community after this event, which I summarize as to get fast "award" through the "fastest finger" approach (i.e., be fast and then you will publish in high-level journals),**

**but still this is not justifying a lack of the survey and discussion on the existing findings. In respect to that, the title should also be changed, to avoid "grand" words, so be more precise with respect of the manuscript content.**

*We agree that some researchers took the "fast as possible" approach, and in our recent literature review, we noticed at least three papers which were submitted within two weeks after the eruption.*

*We hope that our detailed attention to the water level impacts from the atmospheric shockwaves(s), our discussion of energy decay and comparisons to other recent tsunami events, and our focus on the amplification factor, β, will be novel contributions to the discussion about the Tonga event, which will likely take years to fully understand.*

*We also agree with your sentiment that the title should be changed. We now have modified the title to be more descriptive of what we actually did in the paper:*

*"Global water level variability observed after the Hunga Tonga-Hunga Ha'apai volcanic tsunami of 2022".*

*Reviewer #2 Comments:*

**This is yet another article about the atmospheric waves and subsequent ocean waves observed after the explosive eruption of the Hunga Tonga-Hunga Ha'apai volcano. Overall, it is an interesting study with detailed analyses of all available observations (i.e., following the open science policies) during and after the volcano explosion. However, the authors tend to oversell their results: grand statements (e.g., title), movies not providing any information used in the article, lack of reference to other studies related to their results, renaming physical processes already fully documented in the literature (e.g., Lamb waves), etc. Consequently, I recommend major revisions following my comments below.**

*Thank you for your effort in reading our paper. It was indeed our main motivation to analyze all available open-source data and explore the impacts to global water levels due to the combination of "meteo" and marine mechanisms, and to discuss the impacts of the repeated atmospheric component.*

*Regarding the "grand-sounding" title. In the new version, we have renamed the title to: "Global water level variability observed after the Hunga Tonga-Hunga Ha'apai volcanic tsunami of 2022"*

*As to missing references to some other studies, this event has attracted a lot of attention, and papers are coming out at a rapid clip, even since our submission to Ocean Science (Sept of 2022). It has thus been challenging to "keep up" with all the papers, and we could not have referenced those that appeared after our submission. We have now added the papers you have mentioned, as well as many others we have found since receiving this review. Most of the new references are included in the Introduction section (which has been heavily restructured) in the introduction. We have also added more background on weather-related meteotsunamis in Section 2 about the Physics of atmospheric tsunamis and made some contrasts between the two types of meteotsunamis. We have also included some of these works in our Discussion and compared our observations and results to those of other researchers.*

*Finally, we have changed our terminology to better describe the mechanisms and effects observed and reported here.*

*More detailed responses are given to your specific comments below.*

**Major comments:**
**1. I appreciate the efforts made by the authors to find names for physical processes not named before. However, I have several objections to the way this naming was done in the article. First, Lamb waves are already defined and documented and their name should be kept. The authors cannot single handily decide to rename this process named after Lamb research in 1911.**

*We use "Lamb waves" now in the revision.*

**Second, "air-shock" seems to me a poor choice as directly invoking (at least to me) "air-shock absorbers" and not any physical process. I propose "sonic boom" related atmospheric waves (which is based on the physical process occurring and heard during the explosion). I am not imposing this name as I am sure some objections can be easily voiced against it. However, I would like to engage the authors to rethink more deeply of the names they give. Third, these "sonic boom" related atmospheric waves (or whatever other name the authors might chose) include, at least, both Lamb waves (barotropic process) and Perkaris waves (3D internal waves) and maybe others not known to me. Fourth, once a name is decided, the authors must stick with it. Is it "air shock", "air-shock", or "atmospheric" tsunami?**

*We struggled with how to name the observed phenomena, as we had trouble finding relevant background and context outside of weather-related meteotsunamis, which are similar to what happened with Tonga, but not the same. We thought of "air-shock" as a shock wave in the atmosphere and not something to do with shock absorbers! We have given this point careful thought and have decided here to use "volcanic meteotsunami". Finally, we thank the reviewer for the information about the Perkaris mode/wave aspect of the eruption. We only recently learned about Perkaris modes and will include more information about this phenomenon in the revised manuscript. Finally, note that both "air-shock" (used as an adjective) and "air shock" (used as a noun) are grammatically correct, so that this term sometimes had a hyphen, and sometimes not.*

**Finally, there is already a name for tsunamis driven by atmospheric forcing: "Meteotsunamis". So why using another name? I suppose one argument is that it is not a "Meteo" event … But, in this case, should the "meteorological ground stations" be also renamed as they measured the mean-sea level pressure from the Lamb and Perkaris waves? I propose to keep meteotsunamis with two categories: "weather" and "sonic boom" related events. This approach keeps the historical naming used for more than 20 years in the scientific community and distinguishes between the atmospheric sources (as generally done in the tsunami community that distinguishes the tsunami sources: earthquake, volcano, landslide, asteroid, etc.).**

*We appreciate your thoughts about terminology and have eliminated the "air-shock" terminology, as well as including several new meteotsunami-related references in our revision. The comment: "But, in this case, should the "meteorological ground stations" be also renamed as they measured the mean-sea level pressure from the Lamb and Perkaris waves?" is not pertinent, because it would also apply to tide gauges, AKA water level stations ( a later, more general term). The names of both meteo and tide stations are traditional, referring to their original purpose – and both do much more than they did originally. The etymology of the word "meteorology" is:* "science of the earth's atmosphere, scientific study of weather and climate," especially with a view to forecasting the weather, 1610s, from French météorologie and directly from Greek meteōrologia "treatise on celestial phenomena," literally "discussion of high things," from meteōron "thing high up" (see meteor) + -logia "treatment of" (see -logy)."

(https://www.etymonline.com/word/meteorology), but also: "The science that deals with the phenomena of the atmosphere, especially weather and weather conditions" and "The science which treats of the motions and phenomena of the earth's atmosphere; the scientific study of weather and climate, their causes, changes, relations, and effects." (https://www.wordnik.com/words/meteorology). *Therefore, subsuming this sort of tsunami under the general heading of "meteotsunami" seems reasonable. We do not accept, however, the "sonic-boom" suggestion, however, because a sonic boom is a sound shock wave, travelling at the speed of sound. Lamb waves travel more slowly than the speed of sound, and a sonic boom was heard in parts of Alaska before the meteotsunami arrived. We suggest, therefore, the terminology of "volcanic meteotsunami", but not "sonic-boom meteotsunami".*

**More on the naming, why using "Tonga-Hunga-Ha'apai" when the entire scientific community as well as the site the authors refer to (https://volcano.si.edu; line 54 of the article) name the volcano: Hunga Tonga-Hunga Ha'apai. I have no problem to simplify the name to "Tonga event" (and not Tonga-Event; line 483) in the rest of the article but the first mention of the volcano should correspond to the accepted name (to my understanding, the full provided name is not controversial).**

*We hope to follow naming conventions that are most respectful to the nations involved, so we have no problem with using the correct official name.  After the first mention, we will consistently use the "Tonga event" in the reminder of the paper.*

**2. Title: I personally do not see the "lessons" learnt from this event in this specific article. The fact that warning systems should include "sonic boom" related (or whatever name you wish to give them) meteotsunamis is already published (https://doi.org/10.1175/BAMS-D-22-0164.1; https://doi.org/10.1002/essoar.10511565.1). Interestingly, the inference of the authors (made with the analysis of the Tonga event observations) that such meteotsunamis (free waves + force waves) can reach 10 m has been demonstrated with numerical models. This leads to my third point.**

*We have elected to change the title of this paper to be:  "Global water level variability observed after the Hunga Tonga-Hunga Ha'apai volcanic tsunami of 2022". However, the reference provided above, while interesting and cited in our revised version, is to a submitted manuscript that had not been peer reviewed at the time and was not published until Jan 2023. Thus, there is no reason we should have known about or cited this item.*

**3. The authors fail to cite a vast majority of the already published literature. I understand that their article is not a review of all the work already done but some "important" references (in the sense of what was already found similar/different to their study) seemed to have be ignored. I encourage the authors to revise their literature review as something cannot be presented as a new result/conclusion if it already has been published. For example, I think the authors should get familiar with the work of Okal and Synolakys (2016) which is nicely related to their free wave analysis and might (or not, I encourage the authors to look at it in depth) be related to the behavior in the Caribbean. Amplification of the meteotsunami waves nearshore depending on the "amplification factor" of given harbors or bays is also a well-known and well-documented characteristic (please review the meteotsunami literature), this might be another reason for the Caribbean response (again to be proven or rejected by the authors).**

*We appreciate your suggested references. This study from 2016 is pertinent. The Caribbean results were, to us, the most puzzling, and we have added more materials and suggestions about what may have happened there. We have also added a new paragraph to the introduction to discuss new studies and another in the Discussion, where we compare our results to those of other papers.*

**4. Theories and methodologies presented in the article are what I would call "classic" and, in my opinion, the article tries too hard to sell facts that are already partially known from other studies (on the Tonga event itself but also from the tsunami and meteotsunami communities). Putting together all the pieces of a puzzle is good enough achievement for a scientific article, no need to amplify the novelty and the reach of the results. For example, (1) the additional movies do not add any value to the article, (2) the lengthy description of the Garret (1976) theory does not particularly add to the analyses (reference and brief summary should be enough), (3) the title is "sexy" but not really related to the true findings of the study, etc. However, last figure summarizing all the ocean processes is, for me, the main achievement of the article (finalized puzzle) and could be even improved by better representing the synchronization of the events (e.g., free wave occurs after the forced wave) and being more specific with the shallow water processes (e.g., shoaling, harbor resonance, etc.) which play a really significant role as documented for weather related meteotsunami events.**

*The other reviewer had a similar comment, and our response here is similar to those responses. We understand that the theory of Lamb resonance has been described previously, especially in the meteotsunami literature. We include the basic theory to provide a background and foundation for our definition of one of the parameters determined here, the "amplification factor", β. We believe this parameter is a novel part of our study, and that a discussion of the fundamentals helps provide appropriate context for our amplification factor. The amplification factor is also the main focus of our final figure. Without a suitable treatment of the foundational theory, the meaning of our amplification factor would be obscured. However, we have carefully looked at this part of the paper and tried to tighten up the text to only provide the most important background.*

*Regarding your other comments:*

*(1) The movies have been omitted in this version.*

*(2) As stated above, the background theory of Garret (and others) is included since it helps provide context for our determination of our amplification factor which we denote β.*

*(3) We have changed the title.*

*Finally, we appreciate that you liked our final figure, which we also believe is one of the "best parts" of our paper. We have added a little bit more discussion with this figure to include potential effects of shallow-water processes and the timing of the free/forced waves.*

---

## Author Response (AR2)

**Public justification (visible to the public if the article is accepted and published)**:

Dear authors,

thank-you for your updated manuscript and response to the reviewers. Of course it's difficult to cite papers published after yours is submitted! But thank-you for updating them anyway. And I absolutely agree about fastest-finger-first publishing.

I just have a few technical corrections:

Fig 1 - please consider using the same colour scale for the two time panels.

*Done!*

All figures, please consider your choice of colormap very carefully. I leave it to your judgement, personally I like rainbow colormaps, and they are ok for figures like fig 3e, where the shape clarifies meaning. But in eg fig 2c the red and green dots in the Pacific may be hard to distinguish for some readers.

*Thanks for the advice and suggestions. We elected here to change the background of the maps to make the dots more readable. I wanted to use the full spectrum on this plot, so I hope it is distinct enough to read now.*

Fig 3 - please add a legend to the figures to assist colour assessibility (ie don't rely on the reader being able to identify orange & magenta named in the caption). Nice figure otherwise.
Same for fig S1, and please check all the others.

*We have added some legends in Figure 3 and Figure S1, and I also decided to change the second thin line in 3a to be green instead of magenta, which appeared too close to orange.*

Typesetting - several places discussing amplification, 7[Capital X] is not the right symbol for $7\times$. Anyway, as \beta is a ratio you don't really need the \times.

*Great, thanks again for catching that. I think I found and removed all the unneeded "X"s from the manuscript now. Please let me know if I missed any.*

Please indicate any placenames used on a map somewhere in the paper wherever possible. Eg Kushimoto could be added to fig 8a.

*That is a good idea, thank you! Most plots have too much going on to add too many labels, but I had some room on Figure 8 to add place names for all examples given in*

*Figures 3, 4, 5 (minus the Chilean station not visible on these maps), and Figure 6.*

Fig 5 scalograms fonts have lost quality. Also (all scalogram figures) I'm not sure why the window for long periods is different between different scalograms. It might be an artefact of available data but just check you've got the hours labelled correctly.

*Thanks for noticing this. However, I can't notice any difference between the scalogram plots in Figure 5 from the others, though I admit the fontsize is a bit small. I also double-checked all of these plots, and the hours are all standardized with reference to the actual eruption of the volcano.  Some differences might be seen for tide gauge analyses and buoy analyses (which only recorded at different times). . In any case, I have higher-resolution figure files that I will be submitting with this revision.    Hopefully this will be clearer.*

Fig 6 - I like the way you've expanded each key wave in sub panels. Even though it's small figure it's really clear. I might use that!

*Thanks!  This took quite a long time to get right!*

Section 3.3 - I think from the Appendix that when you say "detide" you mean "remove all low frequencies"? The caption for S3 should say this, otherwise the residual includes storm surges that happened the same day, etc.

*Yes, that is the better way of saying it.  I have made changes where required.*

Please ensure all references have dois.

*I have spent a lot of time finding all DOIs that exist. Some of the citations do not have any DOIs to find no matter how much I search, like the Green paper from 1838, and the Press paper from 1956, and the Shufeldt note from 1888.  Then, a few of my citations are books, which I believe I reported correctly. Everything else should have a DOI*

I noticed that your figures 1, 2, 8 and some figures in the Supplement contain maps. For the next revision, I kindly ask you to clarify if you have created the maps or were they created by a map provider? If the maps were not created by you, please provide in your revised file that the copyright is denoted in the figure itself. If this is not possible, please provide it in the caption. If you are the originator, you can simply inform us. Please see the section "Manuscript composition" in our manuscript preparation guidelines:
https://publications.copernicus.org/for_authors/manuscript_preparation.html

*I already discussed this point with you in a previous email. All map figures now have a short statement saying: "Maps made in MATLAB using data from Natural Earth."*